# Regional and global temperature response to anthropogenic SO₂ emissions from China in three climate models

**M. Kasoar[1], A. Voulgarakis[1], J.-F. Lamarque[2], D. T. Shindell[3], N. Bellouin[4], W. J. Collins[4], G. Faluvegi[5], and K. Tsigaridis[5]**

[1]{Department of Physics, Imperial College London, London, UK}

[2]{NCAR Earth System Laboratory, National Center for Atmospheric Research, Boulder, CO, USA}

[3]{Nicholas School of the Environment, Duke University, Durham, NC, USA}

[4]{Department of Meteorology, University of Reading, Reading, UK}

[5]{Center for Climate Systems Research, Columbia University, and NASA Goddard Institute for Space Studies, New York, NY, USA}

Correspondence to: M. Kasoar (m.kasoar12@imperial.ac.uk)

**Abstract**

We use the HadGEM3-GA4, CESM1, and GISS ModelE2 climate models to investigate the global and regional aerosol burden, radiative flux, and surface temperature responses to removing anthropogenic sulfur dioxide (SO₂) emissions from China. We find that the models differ by up to a factor of six in the simulated change in aerosol optical depth (AOD) and shortwave radiative flux over China that results from reduced sulfate aerosol, leading to a large range of magnitudes in the regional and global temperature responses. Two of the three models simulate a near-ubiquitous hemispheric warming due to the regional SO₂ removal, with similarities in the local and remote pattern of response, but overall with a substantially different magnitude. The third model simulates almost no significant temperature response. We attribute the discrepancies in the response to a combination of substantial differences in the chemical conversion of SO₂ to sulfate, translation of sulfate mass into AOD, cloud radiative interactions, and differences in the radiative forcing efficiency of sulfate aerosol in the models. The model

with the strongest response (HadGEM3-GA4) compares best with observations of AOD regionally, however the other two models compare similarly (albeit poorly) and still disagree substantially in their simulated climate response, indicating that total AOD observations are far from sufficient to determine which model response is more plausible. Our results highlight that there remains a large uncertainty in the representation of both aerosol chemistry as well as direct and indirect aerosol radiative effects in current climate models, and reinforces that caution must be applied when interpreting the results of modelling studies of aerosol influences on climate. Model studies that implicate aerosols in climate responses should ideally explore a range of radiative forcing strengths representative of this uncertainty, in addition to thoroughly evaluating the models used against observations.

## 1   Introduction

Short-lived atmospheric pollutants such as aerosols have very inhomogeneous spatial distributions. This means that, unlike long-lived greenhouse gases such as $CO_2$, the radiative forcing due to aerosols is highly variable, and the resulting climate response may be strongly influenced by the region of emission and the prevailing circulation patterns. There is increasing interest in trying to understand how aerosol forcing from different regions affects the climate, both locally and remotely. For example, Shindell and Faluvegi (2009) and Shindell et al. (2012) looked systematically at the response of temperature and precipitation to single-species forcings imposed in different latitude bands, and showed that the influence of remote forcings on certain regions can often outweigh and even have opposite sign to the influence of local forcings. Teng et al. (2012) investigated the global temperature response to drastically increasing carbonaceous aerosols only over Asia, finding a strong remote effect on US summertime temperatures.

One of the most important anthropogenically-sourced aerosol species is sulfate ($SO_4$) (e.g. Myhre et al., 2013b). Sulfate-containing aerosols are formed following chemical conversion of gaseous sulfur dioxide ($SO_2$) emissions from fossil-fuel combustion, as well as natural sources such as volcanic $SO_2$ and ocean dimethyl sulfide (DMS) emissions (e.g. Andres and Kasgnoc, 1998; Andreae and Crutzen, 1997). Sulfate particles strongly scatter incoming shortwave (SW) radiation, which helps to increase the planetary albedo and cool the surface. They also act as cloud condensation nuclei, leading to additional cloud droplets forming in supersaturated conditions, which increases cloud albedo and again cools the Earth system

(Boucher et al., 2013). Historically, cooling from sulfate aerosol, predominantly in the more industrialised northern hemisphere, has been implicated by a range of modelling studies in disrupting climate since the mid-20th century. For instance, Booth et al. (2012), Hwang et al. (2013), and Wilcox et al. (2013) discussed the importance of historical aerosol cooling in modulating large-scale temperature and precipitation patterns, while other studies such as Bollasina et al. (2011), Dong et al. (2014), and Polson et al. (2014) have looked at the impact of historical aerosols on regional climate features such as the monsoon systems or Sahelian rainfall.

The few studies that have investigated specific regional aerosol forcings (e.g. Shindell and Faluvegi, 2009; Shindell et al., 2012; Teng et al., 2012) typically used a single climate model at a time to investigate the climate response to idealised, historical, or projected forcings. However models vary considerably in their representation of aerosols and their radiative properties, resulting in a large uncertainty in aerosol radiative forcing (e.g. Myhre et al., 2013b; Shindell et al., 2013a). When investigating the climate response to regional aerosol emissions, such uncertainties are likely to be confounded even further by the variability between models in regional climate and circulation patterns, and variation in the global and regional climate sensitivity (the amount of simulated warming per unit radiative forcing). To best interpret the findings of single-model experiments with regional aerosol forcings, it is therefore critical to understand the range of uncertainty in the climate response that may arise as a result of structural and parametric differences between climate models.

We investigate here the range of variability that can arise in the translation of a regional emission perturbation to a climate (temperature) response, between three different state-of-the-art global climate models. We select as a case study the removal of $SO_2$ anthropogenic emissions from the region of China. Since China is currently the largest anthropogenic source region of sulfur dioxide (Smith et al., 2011) and hence anthropogenic aerosol, this regional perturbation represents a substantial modification to global aerosol levels, with the additional characteristic of being localised over a particular part of the world. This aspect of our experiment is distinct from many previous model intercomparison studies, which have typically compared the climate response in models forced by global historical trends in aerosols (for example, Shindell et al., 2015; Wilcox et al., 2013), or which have only considered the impact of regional emissions on long-range pollution transport and on radiative forcing (for example the HTAP and AeroCom experiments (HTAP, 2010; Yu et al., 2013; Kinne et al., 2006; Schulz

et al., 2006; Textor et al., 2006; Myhre et al., 2013)), but have not investigated the range of model climate responses to a regionally localised emission perturbation. The potential importance of remote climate effects due to the strong zonal asymmetry created by such regional emissions has therefore not yet been explored in multi-model studies. Single-model studies such as Teng et al. (2012) suggest though that regionally localised forcings can produce significant climate teleconnections in at least the longitudinal direction.

In the following sections we first describe the three models employed, and our experimental setup (Sect. 2). We then present the results of the radiative flux and surface temperature responses to the removal of Chinese $SO_2$ (Sect. 3), and analyse the possible reasons for differences between the model responses (Sect. 4). Finally, in Sect. 5 we present our conclusions.

## 2 Model descriptions and experimental set-up

The three models we employ are the Hadley Centre Global Environment Model 3 – Global Atmosphere 4.0 (HadGEM3-GA4), the Community Earth System Model 1 (CESM1), and the Goddard Institute for Space Studies ModelE2 (GISS-E2). To allow the climate system to freely respond, the models are all used in a fully coupled atmosphere-ocean configuration. These three models all feature explicit aerosol modelling, and include both direct and indirect radiative effects of aerosols. However the models all vary in the details of the parameterisations used, the dynamical cores, radiation and cloud schemes, model grids and horizontal and vertical resolutions, land surface and ocean components, etc. This lack of common structural features makes these three models well suited to contrast against one another and probe the range of potential model uncertainty, as we do here. The models are briefly described below, and the key references and features are summarised in Table 1.

### 2.1 Model descriptions

### 2.1.1 HadGEM3-GA4

For HadGEM3, we use the Global Atmosphere 4.0 version of the model (Walters et al., 2014) in a standard climate configuration with a horizontal resolution of 1.875° longitude x 1.25° latitude in the atmosphere, with 85 vertical levels and the model top at 85km. The atmosphere

is coupled to the JULES land surface model (Walters et al., 2014). Here we prescribe fixed vegetation and also globally-uniform observed mass-mixing ratios for $CO_2$, $CH_4$, and other long-lived greenhouse gases, taking their year-2000 values from the CMIP5 historical dataset (Meinshausen et al., 2011). A zonally-uniform present-day ozone climatology is also prescribed in the radiation scheme, derived from the SPARC dataset (Cionni et al., 2011). The atmospheric model is coupled to the NEMO dynamical ocean model (Madec, 2008) and CICE sea-ice model (Hunke and Lipscombe, 2008), which are run with a 1° horizontal resolution, and 75 vertical depth levels for the ocean.

HadGEM3-GA4 can be run with a choice of two aerosol schemes of differing complexity – CLASSIC (Bellouin et al., 2011), and GLOMAP (Mann et al., 2010). Here we use the simpler CLASSIC scheme, which is less computationally expensive, and is also the aerosol scheme that was used for CMIP5 simulations with the predecessor of this model (HadGEM2). CLASSIC is a mass-based scheme, meaning that only aerosol mass (and not particle number) is tracked, and therefore all aerosol species are assumed to be externally mixed. The scheme includes an interactive representation of sulfate in three modes (Aitken, accumulation, and in-cloud), fossil-fuel black carbon, fossil-fuel organic carbon, and biomass-burning aerosol in three modes (fresh, aged, and in-cloud), dust in six size bins, and sea-salt in two modes (jet and film), as well as an offline biogenic aerosol climatology. The scheme can also include a representation of nitrate aerosol, but this option was not used here.

The sulfate component of the scheme (Jones et al., 2001) includes tracers for two gas-phase precursors: $SO_2$ from anthropogenic and natural sources, and DMS from natural sources. These are emitted into the atmosphere and can undergo advection, wet and dry deposition, or oxidation using prescribed 4D oxidant fields (Derwent et al., 2003). In CLASSIC, oxidation of $SO_2$ to $SO_4$ aerosol can proceed through three possible reaction pathways: in the gas phase by reaction with OH, or in the aqueous phase by reaction with either $H_2O_2$ or $O_3$.

The radiative transfer scheme of Edwards and Slingo (1996) is used with six spectral bands in the shortwave, and all aerosol species interact with radiation. The hygroscopic aerosols (sulfate, organic carbon, biomass-burning aerosol, sea-salt) can also interact with clouds via their role as cloud condensation nuclei. Cloud droplet number concentration and effective radius are determined from the mass concentration of these aerosols, which affects the simulated cloud lifetime (2nd indirect effect) and cloud brightness (1st indirect effect) as described in Bellouin et al. (2011) and Jones et al. (2001).

## 2.1.2 CESM1

CESM1 is run in its standard CAM5-Chem configuration (Tilmes et al., 2015) with a horizontal resolution of 2.5° longitude x 1.875° latitude, and 30 vertical levels, with the model top at approximately 40 km. The atmosphere is coupled to the Community Land Atmosphere version 4 land surface model (Lawrence et al., 2011). In the present configuration, the vegetation distribution is fixed at its 2005 distribution and the $CO_2$ concentration is specified. The atmosphere model is coupled to the POP2 ocean and CICE4 sea-ice models, with an equivalent resolution of 1°.

In the present CAM5-Chem configuration (Tilmes et al., 2015) we use an online representation of tropospheric and stratospheric chemistry so that no chemical constituents are specified, other than specifying the long-lived greenhouse gases' concentrations in the surface layer. CAM5-Chem uses the MAM3 modal aerosol scheme (Liu et al., 2012), which is the same as used for the CESM1 submission to CMIP5. Both aerosol mass and particle number are prognostic, and the scheme simulates sulfate, black carbon, primary organic matter, secondary organic aerosol, dust, and sea salt aerosol species as an internal mixture in Aitken, accumulation, and coarse modes.

The model includes emissions of natural and anthropogenic $SO_2$ and natural DMS as sulfate precursors, and the gas-phase chemistry is coupled to the MAM3 aerosol scheme so that the rate of formation of sulfate aerosols is dependent on the chemical state of the atmosphere. $SO_2$ can be converted to $SO_4$ through three oxidation pathways: by OH in the gas phase, or by either $H_2O_2$ or $O_3$ in the aqueous phase. In addition, the surface area of the prognostic tropospheric aerosols is used to compute heterogeneous reaction rates that affect gas-phase chemistry.

Shortwave radiative transfer is based on the RRTM_SW scheme (Clough et al., 2005) with 14 spectral bands, and aerosols interact with climate through both absorption and scattering of radiation. Aerosol-cloud interactions allow for the effect of aerosols on both cloud droplet number and mass concentrations (Tilmes et al, 2015).

## 2.1.3 GISS-E2

GISS-E2 is run in the configuration used for CMIP5 with a horizontal resolution of 2.5° longitude x 2° latitude, and 40 vertical levels, with the model top at 0.1 hPa (80 km). The atmospheric model is coupled to the dynamic Russell ocean model with horizontal resolution of 1° latitude x 1.25° longitude, and 32 vertical levels as described in Schmidt et al. (2014) and Russell et al. (1995).

Well-mixed greenhouse gases are prescribed as described in Miller et al. (2014), but methane is only prescribed at the surface and is otherwise interactive with the chemistry. The ozone distribution is prognostic throughout the simulated atmosphere, and the chemical mechanism is described in Shindell et al. (2013b). In general, other atmospheric gas and aerosol constituents are also simulated online and interact with each other (via oxidants in both the gas and aqueous phases, heterogeneous chemistry, aerosol-influenced gas photolysis, and secondary coating of dust) and with climate (via radiative effects of ozone and methane, water vapour change due to chemistry, and aerosol direct and indirect effects) in a manner consistent with the physics of the rest of the GCM as described in Sect. 2 of Schmidt et al. (2014).

GISS-E2 has a choice of three aerosol schemes of varying complexity – OMA (Koch et al., 2011; 2006), MATRIX (Bauer et al, 2008), and TOMAS (Lee and Adams, 2012). Following the GISS-E2 CMIP5 configuration, we use here simpler mass-based OMA scheme, which includes sulfate, nitrate, elemental and organic carbon, along with secondary organic aerosols and natural sea-salt and mineral dust. Aerosols are parameterised as an external mixture of dry and dissolved aerosol, with particle size parameterised as a function of relative humidity (Schmidt et al., 2006). The sulfur scheme includes natural emissions of DMS, and natural and anthropogenic emissions of $SO_2$. $SO_2$ from these sources can be oxidised to $SO_4$ aerosol through two reaction pathways: by OH in the gas phase, or by $H_2O_2$ in the aqueous phase.

Aerosol direct effects are calculated following the Hansen et al. (1983) radiation model, with six spectral bands in the shortwave. Aerosol indirect effects are calculated as described in Menon et al. (2010), such that cloud droplet number concentration and autoconversion rate depend on the local concentration of aerosol.

## 2.2   Experimental setup

For this study we investigate the surface temperature response to an idealised regional emission perturbation, on a centennial timescale. Each model has a control simulation, initialised from

a present-day state, which is forced with the same anthropogenic emissions of aerosols and their precursors following the year-2000 ACCMIP emission inventory (Lamarque et al., 2010). The control simulations are run for 200 years with continuous year-2000 conditions. For each model, we then also run a 200-year perturbation simulation from the same initial state, in which $SO_2$ emissions from energy production, industry, transport, domestic use, and waste, are set to zero over the region of China, defined here to be the rectangular domain 80°-120°E, 20°-50°N. These emission sectors contribute 98.7% of the anthropogenic $SO_2$ emitted from this region, so this corresponds to a near complete removal of $SO_2$ emissions from this highly polluting area of the globe. Quantitatively, this perturbation reduces global anthropogenic $SO_2$ emissions from around 104 Tg yr$^{-1}$ to 86 Tg yr$^{-1}$, a reduction of around 17 Tg yr$^{-1}$, or 16.5%.

Additionally, shorter atmosphere-only simulations were performed with HadGEM3-GA4 (identical in setup except that sea-surface temperatures (SSTs) and sea-ice cover are prescribed to year-2000 values) in order to diagnose the effective radiative forcing, as well as the $SO_2$ oxidation rates and $SO_4$ wet deposition rates for this model, referred to in Section 3, Section 4.1, and Section 4.1.1. In CESM1, the $SO_2$ burden, surface $SO_4$ concentration, clear-sky radiative flux, and cloud cover referred to in Sections 4.1.1, 4.2, and 4.3, were all diagnosed from a 30-year extension of the control and perturbation coupled simulations, rather than from the original 200 years.

## 3   Radiative forcing and climate response

We investigate the change in the mean state of the models by taking averages over the last 150 years of the 200-year-long simulations (the first 50 years are discarded to allow the response to the perturbation to establish itself), and taking the difference between the perturbation simulation and the control simulation. As well as plotting maps of 2D variables, we also calculate area-weighted means of temperature, short-wave radiative flux, and aerosol optical depth, both globally and for an east China region (E. China) defined as 100°-120°E, 20°-40°N. This region is found to contain the most intense changes in sulfate aerosol in all three models, and is used from here on to quantify the magnitude of local changes over China. The global- and regionally-averaged quantities, with associated uncertainties where available, are tabulated in Table 2, along with the total sulfate burdens over the globe and E. China, and the ratios of AOD to sulfate burden and SW flux to AOD changes.

The anticipated immediate consequence of removing $SO_2$ emissions from China is that there

will be a reduction in the amount of sulfate aerosol formed, leading to a positive shortwave

(SW) radiative forcing. Figure 1 shows the changes in net downward top-of-atmosphere (TOA)

SW radiative flux in each of the three models. For HadGEM3-GA4 and GISS-E2, the plot is

stippled in locations where the change exceeds two standard deviations, estimated for

HadGEM3-GA4 from the grid-point standard deviations from six 150-year-long year-2000

control simulations with perturbed atmospheric initial conditions, and for GISS-E2 from 12

non-overlapping 150-year sections of a 1900-year-long pre-industrial control simulation that

had reached radiative equilibrium. Such uncertainty analysis has not been performed for

CESM1 due to lack of the necessary unforced simulation output for the version of the model

used here. For reference, Fig. 1 also shows the outline of the E. China region, which

corresponds well to the region of maximum SW flux changes in all three models.

Figure 1 reveals that there is a very substantial variation between the models in the intensity of

the local radiative flux change over China. GISS-E2 shows a fairly weak increase in net

downward SW flux over E. China, with a local increase (from Table 2) of 0.91 W m$^{-2}$ and an

insignificant global mean change (-0.034 W m$^{-2}$), whereas HadGEM3-GA4 shows a very

pronounced change of 5.3 W m$^{-2}$ locally over E. China, and a global mean value of 0.28 W m$^{-2}$

$^{-2}$. CESM1 lies in the middle, with a moderate local SW flux change of 4.2 W m$^{-2}$, and 0.19 W

m$^{-2}$ in the global mean. Between GISS-E2 and HadGEM3-GA4, there is a 6-fold increase in

the intensity of the local SW radiative flux change over E. China.

Because these are fully coupled simulations, the change in the TOA SW flux does not provide

a measure of the shortwave radiative forcing, since the underlying climate has been allowed to

adjust, potentially allowing feedbacks on clouds, and snow and ice cover. A complementary

pair of atmosphere-only simulations were performed with HadGEM3-GA4 to diagnose the

effective radiative forcing (ERF) – the change in TOA radiative flux when feedbacks due to the

slow response of the ocean are prevented (Andrews et al., 2010). The global SW ERF due to

removing SO2 from China in these fixed-SST simulations is 0.18 W m$^{-2}$, 35% smaller than the

0.28 W m$^{-2}$ change in the fully coupled case. However, locally over the E. China region, the

fixed-SST SW ERF was found to be 4.2 W m$^{-2}$, which is only 21% lower than the 5.3 W m$^{-2}$

value in the fully coupled experiment. Moreover, the spatial map of the SW flux anomaly over

China is very similar between the two experiments (Supplementary fig. S1). At least in

HadGEM3-GA4, over E. China the change in sulfate therefore appears to be the dominant

driver of the change in TOA SW flux, and the local change in SW flux over this region is reasonably representative of the local radiative effect of the sulfate perturbation even in the fully-coupled simulations with this model. The same is less true of the global-mean values because of positive feedback from ice melt in the Arctic, and also some small but widespread changes in cloud cover, which globally add up to a sizeable effect in the coupled simulations (not shown).

Based on the fully coupled simulations, the substantial differences in the intensity of SW flux changes over China ultimately translate to very pronounced differences in the strength of the resulting climate response. Figure 2 shows the change in surface air temperatures between the perturbation and control simulations for each of the three models, clearly demonstrating that temperature effects extend far beyond the more localised radiative effects. Again stippling indicates that the response exceeds the $2\sigma$ level in HadGEM3-GA4 and GISS-E2. The difference between GISS-E2 and HadGEM3-GA4 is particularly striking. Apart from a small warming in parts of eastern China and north-east Europe by around 0.1-0.3 K, there is virtually no coherent temperature response across the rest of the globe in GISS-E2. The global mean temperature change (Table 2) is -0.028 K and is not significant. In contrast HadGEM3-GA4 displays significant warming across almost all of the northern hemisphere, with much larger increases in temperature between 0.4-1 K in many regions, not only in China but also in much of the US, northern Eurasia, and the Arctic. The global mean temperature response is +0.12 K. CESM1 sits again in the middle, with clear warming responses between 0.2-0.5 K over much of eastern Europe, Asia, and the western Pacific. Overall the warming response is still less strong and less widespread than in HadGEM3-GA4, with a global mean warming of +0.054 K.

The spatial pattern of warming over Europe and Asia in CESM1 bears some qualitative similarity though to the pattern over the same region in HadGEM3-GA4, suggesting that there may be a similar mode of global response to heating over eastern China in these models, at least across the Eurasian continent. The dynamical mechanisms through which local aerosol emissions are translated to remote response are beyond the scope of this manuscript though. Whether GISS-E2 would have displayed the same pattern had the radiative forcing over China been stronger is impossible to tell from these results; given the small magnitude of the SW flux change it seems that most of the spatial pattern in the temperature response in GISS-E2 can be attributed to internal variability – the largest changes in temperature seen in this model are in

fact a region of cooling over the north-west Atlantic, which is mostly not significant and appears instead to be the result of particularly large internal variability in this region.

## 4    Exploring drivers of diversity

We investigate the differences between these models that lead to such a large variation in the predicted temperature response.    We explore below a number of possible sources of discrepancy.

### 4.1    Differences in simulated aerosol amounts and aerosol optical depths

We address first the possibility that differences in the aerosol schemes themselves, lead directly to very different aerosol loadings between the models, despite the identical change in $SO_2$ emissions applied.    Figure 3 shows the change in column-integrated $SO_4$ in each model as a result of removing $SO_2$ emissions from China.    The models vary in both the distribution and magnitude of $SO_4$ reductions.    In particular, HadGEM3-GA4 has the reduction in $SO_4$ burden fairly concentrated over China.    CESM1 and GISS-E2 simulate changes in $SO_4$ which extend further downwind from the source region, giving a larger spatial footprint, although CESM1 still has large reductions over China as well.

For GISS-E2 and HadGEM3-GA4, more detailed chemistry diagnostics were available from a 5-year period of a HadGEM3-GA4 atmosphere-only control simulation, and a 5-year period of the GISS-E2 coupled control simulation.    For these two models, the difference in spatial extent of the $SO_4$ field from Chinese $SO_2$ emissions seems to be due to faster conversion of $SO_2$ to $SO_4$ in HadGEM3-GA4, resulting in much more concentrated changes in $SO_4$ close to the source.    The $SO_2$ lifetime is around 1.8 times shorter in HadGEM3-GA4, associated with around 45% higher wet oxidation rates in this model.    This difference is due in part to the inclusion of an additional wet oxidation pathway in HadGEM3-GA4: whereas GISS-E2 only includes wet oxidation of $SO_2$ by $H_2O_2$ (around 730 kg(S) $s^{-1}$ globally integrated), HadGEM3-GA4 includes wet oxidation by both $H_2O_2$ and $O_3$, each of which contribute similarly in this model (around 540 kg(S) $s^{-1}$ and 520 kg(S) $s^{-1}$ respectively).

Globally integrated, HadGEM3-GA4 and GISS-E2 simulate fairly similar reductions in $SO_4$ burden, at -0.070 Tg and -0.077 Tg respectively (Table 2).    This, combined with the more

spread-out $SO_4$ field in GISS-E2, means that locally over eastern China HadGEM3-GA4 has a

much more intense reduction in $SO_4$ burden, with 50% of the global reduction occurring over

E. China in HadGEM3-GA4 (-0.035 Tg), compared with only 21% (-0.016 Tg) in GISS-E2.

CESM1 includes the same oxidation pathways as HadGEM3-GA4, and in fact has a slightly

shorter $SO_2$ lifetime still, and so the differences between these two models have different

origins.  CESM1 in fact simulates almost double the global change in $SO_4$ burden as the other

two models, with -0.136 Tg.  This means that although the $SO_4$ reduction spreads further from

the source in CESM1 than in HadGEM3-GA4, CESM1 still has a similar reduction to

HadGEM3-GA4 locally over E. China as well (-0.039 Tg), which is also evident in Fig. 3.

Given that HadGEM3-GA4 and GISS-E2 simulate a similar global reduction in $SO_4$, it is

surprising that there is such a difference in the magnitude of their climate responses.  Also,

given that CESM1 simulates a much larger global reduction in $SO_4$ than the other two models,

it is similarly surprising that this model does not have the largest response.  A partial

explanation may be found by inspecting the change in total aerosol optical depth (AOD), which

is a more direct measure of the radiative properties of the aerosol column.  Unfortunately, the

AOD diagnosed by the models is not completely equivalent:  HadGEM3-GA4 diagnosed clear-

sky AOD, which is done in this model by calculating the relative humidity in the cloud-free

portion of each grid-box, and using this adjusted humidity to calculate the size of the aerosol

droplets in the optical depth calculation (Bellouin et al., 2007).  However CESM1 uses the

unadjusted grid-box relative humidity to calculate the droplet sizes in its optical depth

calculation, thereby providing an all-sky AOD calculation (Neale et al., 2012).  GISS-E2

diagnosed both all-sky and clear-sky AOD, and unless otherwise stated we compare here its

clear-sky AOD, as it is more directly comparable with satellite retrievals of AOD (Kahn et al.,

2010; Levy et al., 2013).  Figure 4 shows these changes in AOD at the 550nm wavelength for

the three models.

As with the radiative flux change, there is a large range in the magnitude of local AOD

reduction, with E. China AOD reductions ranging from 0.047 in GISS-E2 to 0.287 in

HadGEM3-GA4, i.e. about six times higher (Table 2).  This is comparable to the approximately

6-fold range of SW flux change found over this region.  Globally averaged, HadGEM3-GA4

also has a much larger AOD reduction than GISS-E2; 0.0042 compared with an almost

negligible 0.0003 in GISS-E2, despite these two models having a similar change in global $SO_4$

burden.  The much lower globally-averaged value in GISS is partly due to a very small but quite

zonally-uniform compensating increase in nitrate aerosol (absent in HadGEM3-GA4), which occurs across the northern hemisphere (not shown). However, the global change in sulfate-only optical depth in GISS-E2 is still only half that in HadGEM3-GA4 (not shown), and locally around eastern China we find the increase in nitrate optical depth in GISS-E2 is at least an order of magnitude smaller than the decrease in sulfate optical depth, and so nitrate compensation does not substantially contribute to the discrepancy in local AOD changes. We therefore still find that HadGEM3-GA4 simulates a considerably larger change in sulfate optical depth per unit change in $SO_4$ burden at both global and local scales. Having the largest change in AOD per unit change in aerosol burden (Table 2) appears to be key to this model simulating the largest climate response.

Comparing the clear-sky and all-sky AOD for GISS-E2 (for which we have both diagnostics), we find that the simulated reduction in E. China all-sky AOD (-0.183) is much larger than the reduction in clear-sky AOD (-0.047). We cannot be sure that the same would apply to CESM1, but it suggests that we might expect the all-sky values we have for CESM1 to be larger than the equivalent clear-sky values. Given this, it is all the more surprising to find reductions of all-sky AOD in CESM1 for the E. China region of -0.076 and for the global mean of -0.0013 (Table 2), which lie in between the clear-sky values of GISS-E2 and HadGEM3-GA4 even though CESM1 had the largest change in $SO_4$ burden both locally and globally.

The AOD changes per unit burden change are summarised in Table 2, and it is clear that there is a large diversity between the models. The possible contributors to diversity in the AOD per unit burden are extensive, and a full analysis of them is beyond the scope of this paper. Host model effects, such as different cloud climatologies and radiative transfer schemes, are one likely contributor. Stier et al. (2013) suggests that one third of total diversity originates there. Relative humidity, which drives water uptake (hygroscopic growth), is also diverse among models. For example, Pan et al. (2015) find that over India, boundary-layer RH is the main source of diversity. At the more basic level, assumed composition and hygroscopic growth curves also often differ between models – in this case, the aerosol scheme used for HadGEM3-GA4 assumes that all sulfate is in the form of ammonium sulfate, whereas CESM1 and GISS-E2 both assume a mixture of ammonium sulfate and sulfuric acid, and additionally all three models use different sources for their hygroscopic growth parameterisations (Bellouin et al., 2011; Liu et al., 2012; Koch et al., 2011; and references therein).

The changes in SW radiative flux and the final climate response seem to correlate with the changes in AOD much better than with the changes in SO$_4$ burden for HadGEM3-GA4 and GISS-E2, where over China there is a 6-fold difference both in AOD and in SW flux change between these two models. For CESM1, the all-sky AOD change over E. China is about 1.6 times larger than the clear-sky change in GISS-E2 (Table 2). If we used instead all-sky AOD from GISS-E2 (not shown in Table 2), we find that the AOD change over E. China is more than 2 times smaller in CESM1 than in GISS-E2. However, the change in TOA SW over the same region is about 4.7 times larger in CESM1, and so it seems that unlike the discrepancies between HadGEM3-GA4 and GISS-E2, differences in the AOD response cannot explain the difference in the magnitudes of radiative flux change between CESM1 and GISS-E2 (see Sect. 4.2).

### 4.1.1 Validation of aerosol fields

To get an indication of whether the model-simulated AODs are realistic in the region of interest, we compare the mean AOD from each model's control run with station observations in Asia from the AERONET radiometer network (Holben et al., 2001). Because of the limited number of stations in the region with long data records, we use the observed AOD at 500 nm from all AERONET stations able to provide an annual mean estimate for at least one year, averaged over all years for which an annual mean was available, (generally ranging between 1998 and 2014 in different stations), and compare this with the annual mean AODs at 550 nm from the three models, masked to the locations of the AERONET stations (Supplementary fig. S2). Focusing on stations in E. China (eight in total), we find that HadGEM3-GA4 compares best with AERONET in this region with a mean station bias of -22%, whilst both GISS-E2 and CESM1 appear to be biased lower in this part of the world, with mean biases of -56% and -60% respectively.

We also calculate the area-weighted mean AOD as observed by the MODIS and MISR satellite instruments. The MODIS (Moderate Resolution Imaging Spectroradiometer) instrument is flown on both the Terra and Aqua satellites, whilst MISR (Multi-angle Imaging SpectroRadiometer) is flown on Terra. For MODIS we use the collection 6 combined Deep Blue + Dark Target monthly AOD product at 550 nm (Levy et al., 2013) (available from https://ladsweb.nascom.nasa.gov/), averaged from both Terra and Aqua satellites, and take a 10-year average from 2003-2012 (2003 being the earliest year that data from both satellites is

available). For MISR we use the best estimate monthly AOD product (Kahn et al., 2010) version 31 (available from https://eosweb.larc.nasa.gov/) at 550 nm over a 15-year averaging period, from 2000-2014 (2000 being the earliest year MISR data is available). For MODIS the area-weighted E. China mean AOD is 0.51, whilst for MISR it is 0.31, so regionally there is a considerable uncertainty in these observations. HadGEM3-GA4 overestimates the AOD compared with both instruments (though only slightly so when compared to MODIS), with a regional average AOD of 0.58, whilst GISS-E2 and CESM1 underestimate it with regionally-averaged AODs of 0.23 for both models. Globally the two instruments are in better agreement, with MODIS giving a global average AOD of 0.17 and MISR giving 0.15. Again HadGEM3-GA4 overestimates global AOD compared with both instruments (0.22) whilst GISS-E2 and CESM1 both underestimate it (0.13 and 0.12). Given that CESM1 diagnosed all-sky AOD, whereas satellite retrievals are only possible for clear-sky conditions, the underestimate for this model is likely greater than these numbers suggest.

There is considerable variation in the observations as well as the models. Globally-averaged, GISS-E2 seems to compare best against MODIS and MISR, though tentatively HadGEM3-GA4 seems to have the more accurate AOD over China, comparing best regionally with both AERONET and MODIS, though poorer against MISR. This suggests that the more concentred sulfate aerosol burden and larger AOD reduction simulated by HadGEM3-GA4 over this region may be more realistic. We note though that since these observations only measure total AOD and cannot differentiate by species, the comparison cannot show for certain that the higher sulfate optical depth specifically is more realistic in HadGEM3-GA4. The AOD reduction over E. China due to removing Chinese $SO_2$ represents 50% of the climatological total AOD in HadGEM3-GA4 over the region, compared with 34% in CESM1 and only 20% in GISS-E2. Even if the total AOD in HadGEM3-GA4 is more realistic, there is still considerable variation between the models as to what fraction of that total AOD is due to Chinese $SO_2$ emissions. This is illustrated further for the two extreme cases, HadGEM3-GA4 and GISS-E2, in Supplementary Fig. S3, which shows that the fraction of climatological AOD made up by sulfate is consistently higher across the east Asian region in HadGEM3-GA4 than in GISS-E2. However, the total non-sulfate AOD is fairly similar across the region in these two models (Supplementary Fig. S4), indicating that the stark difference in the fractional contribution of sulfate comes primarily from HadGEM3-GA4 simulating much greater sulfate AOD alone. Given that regionally GISS-E2 appeared to underestimate total AOD, this would then suggest

that either the higher sulfate AOD in HadGEM3-GA4 is more realistic, or else both models underestimate the non-sulfate AOD.

To try and better constrain whether the sulfate content (rather than total aerosol) is correct, we therefore also compared against the surface sulfate observations conducted in China reported by Zhang et al. (2012) for 2006-2007 (Supplementary fig. S5). However, all three models performed extremely poorly, with HadGEM3-GA4 having a mean bias of -71% (-66% if urban stations are excluded), CESM1 a mean bias of -71% (unchanged if urban stations are excluded), and GISS-E2 a mean bias of -87% (-86% when urban stations are excluded). Although HadGEM3-GA4 and CESM1 are slightly closer to the observed values, the large underestimation despite the relatively good column AOD in HadGEM3-GA4 suggests that at least this model has difficulty representing the vertical profile of sulfate aerosol, and so this comparison with surface measurements may not be particularly useful in constraining the sulfate optical depth or column-integrated burdens. Large underestimations of surface sulfate concentration over East Asia have been reported previously for two other models, MIROC and NICAM, by Goto et al. (2015), suggesting that this is a problem common to many current generation models.

It seems plausible that any differences in the processing of sulfate aerosol would apply to all polluted regions, and not just over China. Indeed, the spatial pattern of the climatological sulfate burden over other major emission regions such as the United States shows a similar characteristic to that over China, with HadGEM3-GA4 and CESM1 having higher burdens close to the emission source regions, whilst GISS-E2 has a more diffuse sulfate distribution (Supplementary fig. S6). With this in mind we also validated the models against surface sulfate observations from the Interagency Monitoring of Protected Visual Environments (IMPROVE) network in the United States (Malm et al., 1994), a dataset with a far more extensive record than the Zhang et al. (2012) dataset for China. Taking 61 IMPROVE stations which have data for at least six years between 1995 and 2005, we find that over the United States all three models are in fact biased high, with GISS-E2 performing relatively better with a mean bias of +10.1%, but HadGEM3-GA4 somewhat worse with +44.5%, and CESM1 worse still with +86%. However, in the case of HadGEM3-GA4 we find that the larger mean bias comes mainly from an incorrect spatial distribution (Supplementary fig. S7), with a high bias on the West Coast but a pronounced low bias in surface $SO_4$ on the East Coast. Consequently, this comparison would suggest that HadGEM3-GA4 in fact has too little sulfate around the principal US emission

regions on the East Coast, even though over that area HadGEM3-GA4 actually has a larger column-integrated sulfate burden (Supplementary fig. S6) and a larger AOD (not shown) than GISS-E2, as was the case for China. This suggests that HadGEM3-GA4 again fails to capture the vertical profile of sulfate, underestimating surface concentrations over this region despite having a high column-integrated burden.

Validation with surface observations therefore seems insufficient to constrain which model performs better with regard to the more climate-relevant column-integrated quantities of sulfate burden and AOD. Returning to Asia, we therefore also tried evaluating the models against column sulphur dioxide observations. We use the gridded, monthly mean Level 3 observations from the Ozone Monitoring Instrument (OMI) (Krotkov et al, 2008) (available from http://disc.sci.gsfc.nasa.gov/Aura) which is flown on the Aura satellite, averaged over eight years from 2005 - 2012. Over the E. China region the mean OMI $SO_2$ is 0.153 Dobson Units (DU), and all three models appear to overestimate this substantially, with very similar regional mean $SO_2$ columns of 0.282 DU for HadGEM3-GA4, 0.272 DU for GISS-E2, and 0.259 DU for CESM1. Spatially, all three models have more diffuse $SO_2$ fields than the OMI observations, in which by contrast the $SO_2$ burden seems much more localised around sources (Supplementary Fig. S8). This may be partly due to the coarse resolution of the models compared with the $0.25°$ satellite product, but also suggests that the lifetimes for $SO_2$ may be too long in all three models, or transport processes too efficient. The surprisingly similar column $SO_2$ burdens in all three models suggests that, at least on regional scales, column $SO_2$ may not constrain $SO_4$ burden that well.

An alternative observational measure which to an extent reflects a column-integrated quantity is the deposition rate, and for the two extreme cases of HadGEM3-GA4 and GISS-E2 we therefore also try comparing against observations of sulfate wet deposition. We use the 3-year mean wet deposition data from 2000-2002 described in Vet et al. (2014) and provided by the World Data Centre for Precipitation Chemistry (http://wdcpc.org, 2014), taking the 6 stations located in China. We exclude the station in Guizhou province in southern China where HadGEM3-GA4 has a bias of +590% and GISS-E2 a bias of +253%. This station only provided data for one year and was flagged as having a high uncertainty in the Vet et al. (2014) dataset; it is also located in a mountainous region and so it could equally be that the models cannot resolve the specific local conditions. Removing this station from the analysis we find for the remaining 5 stations in China that HadGEM3-GA4 performs well with a mean bias of -3.9%,

compared with -64.8% for GISS-E2. This gives an indication that HadGEM3-GA4 has more realistic sulfate deposition directly over China (though the sample size is very small), and supports the earlier findings from the comparison against AERONET and MODIS. If we broaden the analysis to include all stations described as being broadly in Asia – an additional 32 stations – then the mean bias for HadGEM3-GA4 is worsened (-41.8%), whilst the bias in GISS-E2 is slightly improved (-54.1%). HadGEM3-GA4 still performs better over the Asian region as a whole, though less dramatically so (Supplementary fig. S9). This overall picture seems consistent with that of the other observational measures looked at here, although it should be noted that wet deposition rates are dependent not just on the column sulfate burden but also on the amount and distribution of precipitation, and so biases in wet deposition could also be due to incorrect precipitation distribution rather than sulfate.

Still, overall HadGEM3-GA4 seems to compare slightly better than GISS-E2 and CESM1 regionally over E. Asia against observations of total AOD, and better than GISS-E2 regionally against surface sulfate as well as wet deposition observations, although globally and over other regions this model is not necessarily found to compare better in general. This might hint that at least over China, HadGEM3-GA4 has more realistic sulfate optical depth, although none of these comparisons is very conclusive in that respect. Moreover, given that none of these observational measures directly constrains the sulfate radiative forcing, there is also no guarantee that performance with respect to these variables will necessarily translate to a more realistic climate response (see also Section 4.3).

## 4.2    Differences in cloud effects

Sulfate aerosol exerts indirect radiative effects by modifying cloud properties. The strength of these indirect effects is highly uncertain (e.g. Boucher et al., 2013) and differs substantially between the models, having been shown to contribute substantially to inter-model variation in historical aerosol forcing (Wilcox et al., 2015). Differences in the underlying climatologies of the models, particularly with regard to cloud distributions, could also be important. For instance, the radiative effect of sulfate aerosol is modulated by the reflectivity of the underlying surface in the radiation scheme (Chýlek and Coakley, 1974; Chand et al., 2009), which may often be a cloud-top. The low contrast with a highly reflective cloud surface means that sulfate aerosol above a cloud top will have a reduced direct radiative forcing. Blocking of radiation

by clouds will also reduce the direct radiative effects of any aerosols within or below them (e.g. Keil and Haywood, 2003). Additionally, aerosol indirect effects can saturate in regions with a high level of background aerosol (e.g. Verheggen et al., 2007; Carslaw et al., 2013), meaning that the potential for indirect radiative forcing can also vary with the location of clouds. On top of diversity in indirect effects, and in the climatological distribution of clouds, different dynamical changes in cloud cover could also alter the all-sky flux.

In our case, the good correspondence between higher (clear-sky) AOD change in HadGEM3-GA4 and higher (all-sky) SW flux change in this model might suggest that the cloud effects are not the root cause of the larger radiative response in this model. However, the origin of this good correspondence in fact appears to be somewhat dependent on how clouds modify the radiative effects of sulfate aerosol:

For the extreme cases of HadGEM3-GA4 and GISS-E2, comparing the changes in clear-sky TOA SW flux with the all-sky TOA SW flux anomalies (Table 2 and Supplementary Fig. S10) reveals that for clear-sky conditions, there is in fact a much smaller regional discrepancy between these two models: Over the E. Asia region GISS-E2 has a 4.1 $Wm^{-2}$ clear-sky SW flux change, whereas HadGEM3-GA4 has a 5.1 $Wm^{-2}$ flux change. HadGEM3-GA4 still has the larger radiative change, but nowhere near the 6-fold difference that is seen in the all-sky flux (Section 3, and Table 2). This much reduced difference between GISS-E2 and HadGEM3-GA4 in the clear-sky compared with the all-sky anomaly is hard to apportion quantitatively though, because compared with the clear-sky change, the all-sky response incorporates all the contributing factors described above: the additional radiative forcing due to aerosol indirect effects, the screening of direct radiative effects due to clouds blocking radiation and providing a high albedo background, and also any dynamical changes in cloud cover.

In this case, GISS-E2 is found to simulate a small increase in cloudiness in east China due to dynamical changes when sulfate is removed (Supplementary Fig. S11a). Combined with the screening effect of clouds, this appears to almost completely offset the direct forcing of reduced $SO_4$, and results in a far smaller all-sky flux change than clear-sky flux change over E. China (0.9 $Wm^{-2}$ all-sky compared with 4.1 $Wm^{-2}$ clear-sky). HadGEM3-GA4 by contrast has very little difference between all-sky and clear-sky flux changes (5.3 $Wm^{-2}$ and 5.1 $Wm^{-2}$ respectively (Table 2)). The changes in cloud amount over east China are somewhat more mixed (Supplementary Fig. S11c), although area-averaged, the overall cloud change is a small decrease, which should enhance the all-sky flux change. However, spatially as well as in

magnitude the HadGEM3-GA4 all-sky flux change is exceptionally similar to the clear-sky flux change, and does not resemble the pattern of cloud changes (comparing Supplementary Figs. S10e,f, and Fig. S11c), which suggests that changes in aerosol radiative effects are larger than the effect of the small cloud cover changes, and still dominate the all-sky flux changes. Therefore, the very similar regional all-sky and clear-sky SW flux changes in HadGEM3-GA4 imply that unlike in GISS-E2, aerosol indirect effects in HadGEM3-GA4 probably roughly compensated for the presence of clouds reducing the direct effect, so that the change in all-sky combined direct and indirect forcing is similar to the change in clear-sky direct forcing when sulfate is removed.

The picture is different again for CESM1. Comparing the clear-sky and all-sky TOA SW flux changes for this model (Supplementary Figs. S10c,d), we find that regionally, the clear-sky changes are much smaller than the all-sky flux changes – in fact, over China the clear-sky SW flux changes in CESM1 are considerably smaller in magnitude than the clear-sky flux changes in GISS-E2 (comparing Supplementary Figs. S10a,c). Averaged over the E. China region, the clear-sky flux change in CESM1 is only 2.2 $Wm^{-2}$, compared with the 4.1 $Wm^{-2}$ clear-sky change in GISS-E2 (Table 2). However, whereas in GISS-E2 the all-sky SW flux change (0.9 $Wm^{-2}$) was then more than four times smaller than this clear-sky flux change, in CESM1 the all-sky SW flux change is instead almost two times larger than the clear-sky flux change: 4.2 $Wm^{-2}$ regionally averaged.

This is partly again due to cloud changes – in this case CESM1 has predominantly reductions in cloud amount over E. China (Supplementary Fig. S11b), which will have the effect of increasing the all-sky radiative flux change relative to the clear-sky changes. However, as with HadGEM3-GA4, these regional cloud reductions in CESM1 do not match up spatially with the maximum changes in all-sky SW flux seen in Fig. 1b and Supplementary Fig. S10d. Instead, the maximum changes in the all-sky SW flux change match closely the clear-sky SW flux changes (Supplementary Fig. S10c), which in turn correspond very well with the reduction in AOD (Fig. 4b). Both all-sky and clear-sky SW flux changes are maximum around where the AOD reduction is maximum, and in this location the all-sky flux change is still substantially greater than the clear-sky change. This implies that in CESM1 a large aerosol indirect effect, and/or effect of clouds increasing aerosol particle size through hygroscopic growth, overall amplifies the radiative effect of aerosols considerably in cloudy conditions, resulting in the much greater regional change in all-sky flux when aerosol is removed.

Between these three models, then, the way that clouds modify the radiative balance is a major source of diversity over the E. China region in the response to removing $SO_2$ emissions from China. In GISS-E2, the inclusion of clouds greatly reduces the radiative effect of a change in sulfate aerosol. In HadGEM3-GA4, the effect of including clouds is small, and does not change the clear-sky forcing substantially. Finally, in CESM1, including clouds considerably amplifies an otherwise weak clear-sky radiative flux change. We note though that clear-sky diagnostics will be influenced by choices within the models of how aerosol water uptake is determined under the artificial assumption of clear-sky conditions. The all-sky SW flux change, which drives the final climate response, is regionally still the most directly comparable quantity, reflecting the total radiative effect of the aerosol change in the different models.

## 4.3    Differences in aerosol forcing efficiency

An additional source of discrepancy between the models lies in differences in the aerosol radiative forcing efficiency – the direct forcing that results from a given aerosol optical depth or burden (e.g. Samset et al, 2013). A previous model intercomparison looking at radiative forcing as part of the AeroCom Phase II study found that, on a global scale, there was a large variation in the radiative forcing due to aerosol-radiation interactions per unit AOD between different participating models (Myhre et al., 2013a). As a result, whether a model simulates AOD changes correctly, for instance, may not particularly constrain the resultant direct forcing even, let alone the indirect forcing or eventual climate response.

Globally-averaged, the changes in radiative flux and AOD are too small in our experiments to calculate an accurate ratio, but instead we calculate here a regional radiative efficiency by taking the change in clear-sky SW flux over the 100-120E, 20-40N E. China region, and dividing by the AOD change over the same region (Table 2). This is not directly comparable with previous studies like Myhre et al. (2013a), as we use a regionally-averaged number instead of globally-averaged, and for the numerator we use the change in clear-sky TOA SW flux as the best available measure of aerosol direct radiative effect, rather than the direct radiative forcing diagnosed either from double radiation calls or simulations with fixed meteorology. Consequently, we use this metric here mainly to qualitatively highlight differences between the models.

As noted in Sect. 4.1 and 4.2, over the eastern China region HadGEM3-GA4 has a 6-fold larger mean AOD reduction (-0.29) compared with GISS-E2 (-0.047), but only slightly larger clear-sky SW change (5.1 W m$^{-2}$ compared with 4.1 W m$^{-2}$). As a result, the regional radiative efficiency for HadGEM3-GA4 is much smaller than that of GISS-E2: -17.6 W m$^{-2}$ compared with -87.2 W m$^{-2}$ per unit AOD change (Table 2). If instead of AOD we normalise by the change in sulfate burden integrated over the same region, we find a similar relationship: HadGEM3-GA4 has a smaller regional mean change in clear-sky SW flux per Tg sulfate than GISS-E2: -145 W m$^{-2}$ Tg$^{-1}$ compared with -256 W m$^{-2}$ Tg$^{-1}$. Proportionally though, the discrepancy is not as great when normalising by change in sulfate burden, due to the much larger AOD per unit mass of sulfate simulated in HadGEM3-GA4. Curiously Myhre et al. (2013a) reported results that were qualitatively the inverse of what we show here, finding that the atmospheric component of GISS ModelE has a smaller sulfate radiative forcing than that of HadGEM2 (HadGEM3's predecessor, with a very similar aerosol scheme) when normalised by AOD, although still larger when normalised by column-integrated sulfate burden. The reason for the discrepancy is not clear, though the aforementioned fact that we calculate our numbers for a specific region means that there may be important local factors. The sulfate-specific forcing efficiencies in Myhre et al. (2013) are calculated relative to all-sky direct radiative effect, and so local differences in vertical profiles and cloud screening may therefore change the relationship – however they also evaluated clear-sky forcing normalised by AOD for all aerosol species combined, and again found HadGEM2 to be higher than GISS ModelE.

CESM1 seems to sit in the middle of the range, with a regional radiative efficiency of -28.4 W m$^{-2}$ per unit AOD change (Table 2) – though again with the caveat that for CESM1, the AOD is an all-sky quantity, whereas the values given for HadGEM3-GA4 and GISS-E2 (-17.6 W m$^{-2}$ and -87.2 W m$^{-2}$ respectively) were calculated using clear-sky AOD. GISS-E2 provided both clear-sky and all-sky AOD diagnostics, and using instead the all-sky AOD change from GISS-E2 gives a smaller value of -22.4 W m$^{-2}$ per unit AOD, which suggests that when compared like-for-like, CESM1 (with -28.4 W m$^{-2}$) may in fact have the greater regional radiative efficiency. More directly comparable between all three models is the regional clear-sky flux change normalised by regional change in sulfate burden, which for CESM1 is -55.4 W m$^{-2}$ Tg$^{-1}$. This is considerably lower than either HadGEM3-GA4 or GISS-E2, and indicates that despite having at least average radiative efficiency per unit AOD, the very small translation of sulfate burden to AOD in CESM1 is a dominant factor which prevents this model from simulating a larger SW flux change and climate response than it already does. As noted in the previous

Section though, this small clear-sky flux change per unit sulfate change is compensated by a large indirect effect as well as favourable regional cloud changes, meaning that the all-sky flux change per unit AOD is by far the largest is CESM1 (Table 2), and the all-sky flux change per sulfate burden change is then average in CESM1 (not shown, but readily calculated from Table 2). Similarly, the exceptional reduction in aerosol radiative effects due to clouds in GISS-E2 means that its all-sky flux change per unit AOD is almost exactly the same as that of HadGEM3-GA4 (Table 2), despite the clear-sky regional radiative efficiency being so much larger.

The Myhre et al. (2013a) AeroCom intercomparison found that globally, the atmospheric component of CESM1 (CAM5.1) had a much higher sulfate radiative efficiency than the atmosphere-only version of GISS-E2. In their case, they found CAM5.1 to have approximately 2.25 times higher all-sky direct radiative forcing per unit AOD than GISS-E2. However, the study also found that, globally, the atmospheric component of HadGEM2 had a slightly larger forcing efficiency than CAM5.1 both for sulfate (all-sky) and all aerosols (clear-sky), unlike the somewhat smaller regional efficiencies found here for HadGEM3-GA4 compared with CESM1. Given that our regional values from GISS-E2 and HadGEM3-GA4 also seem to conflict qualitatively with the global values from the AeroCom study, this would suggest that either the global comparison is not relevant on regional scales, or else the radiative efficiency is very sensitive to changes in model configuration and version..

## 4.4   Differences in climate sensitivity

So far we have discussed mainly factors which influence the translation of a change in aerosol precursor emissions to a radiative heating, and these varied strongly between the models. There is a final step in arriving at the climate response, which is the translation of a given radiative heating into a surface temperature change. The climate sensitivity – the amount of warming simulated per unit radiative forcing – is also well known to vary considerably between models, globally (Flato et al., 2013) and regionally (Voulgarakis and Shindell, 2010), and this will additionally impact the strength of the final response. Climate sensitivity is typically estimated from a 2x or 4x global $CO_2$ simulation, giving a large response and a large forcing from which to calculate the ratio. For GISS-E2, a climate sensitivity value of 0.6 K $(W\ m^{-2})^{-1}$ was found in the IPCC AR5 report from a 4x $CO_2$ simulation (Flato et al., 2013) using the regression method of Gregory et al. (2004) to estimate radiative forcing. For CESM1, a value of 1.1 K $(W\ m^{-2})^{-1}$

is obtained from values from a 2x $CO_2$ simulation (Meehl et al., 2013), noting that in this case the radiative forcing was calculated using the stratospheric adjustment method (Hansen et al., 2005). For HadGEM3-GA4, we use a 100-year 2x $CO_2$ simulation that was performed separately as part of the Precipitation Driver Response Model Intercomparison Project (Samset et al., 2016), which gives a value of 1.1 K (W m$^{-2}$)$^{-1}$ based on the Gregory method.

While CESM1 and HadGEM3-GA4 both have very similar climate sensitivities, we see that GISS-E2 has a particularly small sensitivity – in fact, the smallest value of all the CMIP5 models reported in the AR5 report (Flato et al., 2013). This presumably compounds the fact that GISS-E2 simulates the smallest SW flux change of the three models, ensuring that the resulting surface temperature response is comparatively smaller still. Differences in climate sensitivity do not seem to explain any of the variation in the magnitude of the response between CESM1 and HadGEM3-GA4, at least based on these values. However, it is worth noting that the climate sensitivity values that we report are derived from global $CO_2$ forcings, whereas in our case we are looking at the translation of a very regional forcing into a global response. It is not trivial that the global-mean temperature response to a regionally localised forcing is a function only of the resulting globally-averaged forcing, and in particular it may be that different models are more or less sensitive to forcings in specific regions. Unfortunately we know of no study that has calculated climate sensitivity to regional forcings in single or multi-model frameworks. Shindell (2012) calculated climate sensitivities to forcings imposed in different latitudinal bands for the GISS-E2 model, finding that there is considerable variation relative to the global climate sensitivity. In that study, estimates of the response to forcings at different latitudes in three other global climate models, based on the GISS-E2 sensitivities, are found to largely agree to within +/- 20% with the full simulations however, suggesting that regional sensitivities (relative to a model's global sensitivity) may not vary that much between models.

## 5   Conclusions

By applying an identical regional perturbation to anthropogenic $SO_2$ emissions in three different climate models, we observe three markedly different resulting climate responses, ranging from virtually no coherent surface air temperature response in one model (GISS-E2), to pronounced surface warming all across most of the northern hemisphere in another (HadGEM3-GA4). The

third model (CESM1) sits in the middle in terms of both magnitude and spatial extent of the temperature response. This huge variation in climate response corresponds to a similarly large variation in the SW radiative flux change following the reduction in sulfate aerosol. All three models show a fairly localised increase in net downwards SW radiation over China as a result of reduced $SO_2$ emissions from this region, however the magnitude of this radiative heating is substantially greater in HadGEM3-GA4 than in CESM1, which is substantially greater still than in GISS-E2. The response in GISS-E2 is so weak that temperature changes are largely not detectable above the internal variability of the model. The stronger heating in CESM1 and HadGEM3-GA4 produces much more pronounced temperature changes, and even though the radiative heating is localised over China, the temperature responses in these two models are much more spread out, particularly in the zonal direction. This is consistent with the findings of Shindell et al. (2010), who found that the temperature response to inhomogeneous aerosol forcings is more uniform and extends much further from the forcing location in the zonal direction than in the meridional direction.

Comparing the models, we find different $SO_4$ mass changes due to removing $SO_2$ emissions from China, very different ratios of AOD change per mass of sulfate, and very different radiative flux changes per unit AOD change. These differences are compounded further by very large variations in cloud interactions, as well as variations in climate sensitivity, and feedbacks on other aerosol species such as nitrate, which diversify the response further.

Specifically, we find that CESM1 simulates the largest reduction in sulfate burden both globally and locally. HadGEM3-GA4 has the smallest reduction in sulfate burden globally and the second largest reduction regionally, yet it produces by far the largest reduction in AOD both globally and regionally over E. China. Though GISS-E2 and CESM1 both simulate much smaller changes in AOD than HadGEM3-GA4, still the SW flux changes and temperature responses produced are very different between these two models. An inferred larger aerosol-cloud interaction means that CESM1 simulates a particularly large change in all-sky SW flux relative to its fairly small AOD change, so although having a smaller response than HadGEM3-GA4, it is still much closer to it than GISS-E2. In GISS-E2 the clear-sky radiative forcing efficiency of sulfate is very large, but this is almost perfectly compensated for by large reductions in the direct radiative effect of sulfate when clouds are factored in. The absolute AOD change is also much smaller than HadGEM3-GA4 in this model. This then combines with compensating increases in nitrate aerosol globally to reduce the radiative response yet

further, and finally a smaller global climate sensitivity than the other two models results in this

being translated into a largely negligible temperature response.

In addition to differences in the total changes in sulfate and AOD, we find there are also

substantial differences in the spatial distribution of the changes, attributed to differences in the

rate of chemical conversion of $SO_2$ to $SO_4$ which influences how concentrated the aerosol

changes are around the emission region. This implies that even if both the AOD per sulfate

burden and the forcing per unit AOD were identical among the three models, they would still

have different distributions of radiative forcing.

There are no direct observations of sulfate radiative forcing, nor of sulfate optical depth or

vertically-integrated burden, and so we have tried validating the aerosol component of the

models with a range of surface and satellite-based measurements of total aerosol optical depth,

surface sulfate concentration, column $SO_2$, and sulfate wet deposition. All the models have

biases, and no model performs best against all the observational datasets used. Tentatively

HadGEM3-GA4 seems to perform best over China against observations of both total AOD and

sulfate wet deposition, though over some other parts of the world this model performed slightly

poorer, e.g. for global AOD and US surface sulfate concentrations. However, the main

conclusion is that comparison against all existing observational measures is unable to

satisfactorily constrain which model response is more realistic, given that the ratios of both

AOD change per sulfate burden change and SW flux change per AOD (Table 2) are found to

vary so substantially between the models. The model with the largest sulfate mass change

(CESM1) did not have the largest radiative or climate response, and two models with a similar

AOD change (CESM1 and GISS-E2) had markedly different radiative and climate responses.

Given the range of discrepancies that we find in all steps along the conversion of $SO_2$ change

to $SO_4$ change to AOD change to radiative forcing to temperature response, it seems that

knowing how accurate a model is with respect to either sulfate concentrations or total AOD is

far from sufficient to determine whether the climate response to a regional aerosol perturbation

is similarly accurate.

There are several possible avenues for future work to isolate the particular processes that lead

to this model diversity in more detail; for instance studies imposing the aerosol field from one

model into others would remove the diversity introduced by translating emissions into aerosol

concentrations, while imposing surface temperatures and meteorology from one model into

others could remove the diversity introduced by different background climatologies and climate

sensitivities, although this may be difficult practically in complex climate models. A thorough assay of the range of parameter choices and formulae used in the aerosol schemes of various models could also help reveal where assumed aerosol properties diverge. However, without stronger observational constraints on aerosol radiative forcing, it is not clear that this alone could help make models more realistic. In particular, it seems that being able to better constrain not only the column-integrated sulfate burden, but also the AOD per sulfate burden, and the radiative forcing per AOD, would all also be needed. This represents a considerable observational challenge, and until it is possible, the considerable current diversity may be irreducible.

We have only looked here at surface temperature, which is a particularly direct measure of the climate response. The response of other, less well-constrained, climate variables such as precipitation might be expected to show even greater variation. Our results show that there remains a very large uncertainty in current climate models in the translation of aerosol precursor emissions into a climate response, and imply that care must be taken not to over-interpret studies of aerosol-climate interaction if the robustness of results across diverse models cannot be demonstrated.

On a more optimistic note, we remark that in the two models which showed the more substantial change in SW radiative flux (CESM1 and HadGEM3-GA4), both also show a remarkably strong remote temperature response to a relatively localised northern-midlatitude heat source, with qualitatively similar temperature change patterns that extend across much of the hemisphere, indicating that there may be some agreement on the response to a given regional forcing, if not on the forcing itself.

## Data availability

Model output data from all simulations described here is available upon request from the corresponding author.

## Acknowledgements

MK and AV are supported by the Natural Environment Research Council under grant number NE/K500872/1. Also, we wish to thank the European Commission's Marie Curie Actions

International Research Staff Exchange Scheme (IRSES) for funding MK's placement at NASA GISS and Columbia University and facilitating interactions on this work with the US colleagues, as part of the Regional Climate-Air Quality Interactions (REQUA) project. Simulations with GISS-E2 used resources provided by the NASA High-End Computing (HEC) Program through the NASA Center for Climate Simulation (NCCS) at Goddard Space Flight Center. Simulations with HadGEM3-GA4 were performed using the MONSooN system, a collaborative facility supplied under the Joint Weather and Climate Research Programme, which is a strategic partnership between the Met Office and the Natural Environment Research Council. We specifically thank Dr. Fiona O'Connor, Dr. Jeremy Walton, and Mr. Mohit Dalvi from the Met Office for their support with using the HadGEM3-GA4 model.

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

| | HadGEM3-GA4 | CESM1 | GISS-E2 |
|---|---|---|---|
| **Primary model reference** | Walters et al. (2014) | Tilmes et al. (2015) | Schmidt et al. (2014) |
| **Aerosol scheme references** | Bellouin et al. (2011) Jones et al. (2001) | Liu et al. (2012) | Koch et al. (2011) Koch et al. (2006) |
| **Resolution (longitude x latitude)** | 1.875° x 1.25° 85 vertical levels, model top at 85 km | 2.5° x 1.875° 30 vertical levels, model top at 40 km | 2.5° x 2° 40 vertical levels, model top at 80km |
| **Aerosol tracers** | Sulfate, fossil-fuel black carbon, fossil-fuel organic carbon, biomass-burning, dust, sea salt | Sulfate, black carbon, primary organic matter, secondary organic aerosol, dust, sea salt | Sulfate, nitrate, black carbon, organic carbon, secondary organic aerosol, dust, sea salt |
| **Indirect effects included** | Yes (1st and 2nd) | Yes (1st and 2nd) | Yes (1st and 2nd) |
| **$SO_2$ oxidation reactions included** | OH (gas phase) $H_2O_2$, $O_3$ (aqueous phase) | OH (gas phase) $H_2O_2$, $O_3$ (aqueous phase) | OH (gas phase) $H_2O_2$ (aqueous phase) |
| **Chemistry** | Offline (prescribed 4D oxidant fields) | Online | Online |
| **Shortwave radiation** | Edwards and Slingo (1996) 6 spectral bands | Clough et al. (2005) 14 spectral bands | Hansen et al. (1983) 6 spectral bands |

2  Table 1: Key references and features of the three models and their aerosol schemes used in
3  this study

| | | HadGEM3-GA4 | | | GISS-E2 | | | CESM1 | | |
|---|---|---|---|---|---|---|---|---|---|---|
| | | Con | Ch0 | Ch0-Con | Con | Ch0 | Ch0-Con | Con | Ch0 | Ch0-Con |
| Global | Total SO$_2$ (Tg) | 0.637 | 0.592 | -0.045 ± 0.001 | 1.151 | 1.075 | -0.076 | 0.553 | 0.503 | -0.050 |
| | Total SO$_4$ (Tg) | 1.569 | 1.499 | -0.070 ± 0.004 | 1.091 | 1.014 | -0.077 | 1.459 | 1.323 | -0.136 |
| | Mean AOD | 0.217 | 0.213 | -0.0042 ± 0.0004 | 0.131 | 0.131 | -0.0003 | 0.123 | 0.122 | -0.0013 |
| | Clear-sky TOA SW flux (W m$^{-2}$) | 286.0 | 286.2 | 0.184 ± 0.06 | 289.0 | 289.1 | 0.052 | 288.7 | 288.8 | 0.076 |
| | All-sky TOA SW flux (W m$^{-2}$) | 242.3 | 242.6 | 0.279 ± 0.10 | 241.0 | 241.0 | -0.034 ± 0.06 | 236.7 | 236.9 | 0.186 |
| | Mean temperature (K) | 288.6 | 288.7 | 0.115 ± 0.05 | 289.0 | 289.0 | -0.028 ± 0.04 | 288.0 | 288.1 | 0.054 |
| | Δ AOD/ Δ SO$_4$ (Tg$^{-1}$) | | | 0.0603 | | | 0.0042 | | | 0.0094 |
| | Δ Clear-sky SW/ Δ AOD (W m$^{-2}$) | | | -43.8 | | | -173 | | | -58.5 |
| | Δ All-sky SW/ Δ AOD (W m$^{-2}$) | | | -66.4 | | | 106 | | | -145 |
| E. China (100°E-120°E, 20°N-40°N) | Total SO$_2$ (Tg) | 0.035 | 0.006 | -0.029 ± 0.0002 | 0.033 | 0.005 | -0.028 | 0.030 | 0.001 | -0.028 |
| | Total SO$_4$ (Tg) | 0.050 | 0.015 | -0.035 ± 0.0003 | 0.043 | 0.027 | -0.016 | 0.054 | 0.015 | -0.039 |
| | Mean AOD | 0.576 | 0.289 | -0.287 ± 0.002 | 0.232 | 0.185 | -0.047 | 0.227 | 0.151 | -0.076 |
| | Clear-sky TOA SW flux (W m$^{-2}$) | 296.3 | 301.4 | 5.06 ± 0.08 | 294.3 | 298.4 | 4.10 | 305.35 | 307.51 | 2.16 |
| | All-sky TOA SW flux (W m$^{-2}$) | 228.8 | 234.2 | 5.34 ± 0.3 | 233.32 | 234.22 | 0.90 ± 0.3 | 224.16 | 228.36 | 4.20 |
| | Mean temperature (K) | 287.6 | 287.9 | 0.382 ± 0.07 | 288.965 | 289.014 | 0.049 ± 0.07 | 289.110 | 289.404 | 0.294 |
| | Δ AOD/ Δ SO$_4$ (Tg$^{-1}$) | | | 8.23 | | | 2.94 | | | 1.96 |
| | Δ Clear-sky SW/ Δ AOD (W m$^{-2}$) | | | -17.6 | | | -87.2 | | | -28.4 |
| | Δ All-sky SW/ Δ AOD (W m$^{-2}$) | | | -18.6 | | | -19.3 | | | -55.0 |

Table 2: Area-integrated $SO_2$ and $SO_4$ burdens, area-weighted annual means of AOD, net down clear-sky and all-sky TOA SW flux, and surface temperature, and ratios of the changes in AOD to change in $SO_4$ burden, and SW flux to change in AOD, for the globe and the E. China region 100°E - 120°E, 20°N - 40°N. Values are shown for each model for the control simulation (Con), the simulation with no $SO_2$ emissions from China (Ch0), and the difference (Ch0 – Con). AOD is diagnosed for clear-sky conditions in HadGEM3-GA4 and GISS-E2, and for all-sky conditions in CESM1. For models and variables where data was available, error ranges are quoted for the Ch0-Con values and indicate $\pm$ 2 standard deviations, evaluated in HadGEM3-GA4 from an ensemble of six 150-year control runs with perturbed initial conditions, and in GISS-E2 from twelve 150-year segments of a long pre-industrial control run. Values quoted without error ranges indicate that uncertainty was not evaluated.

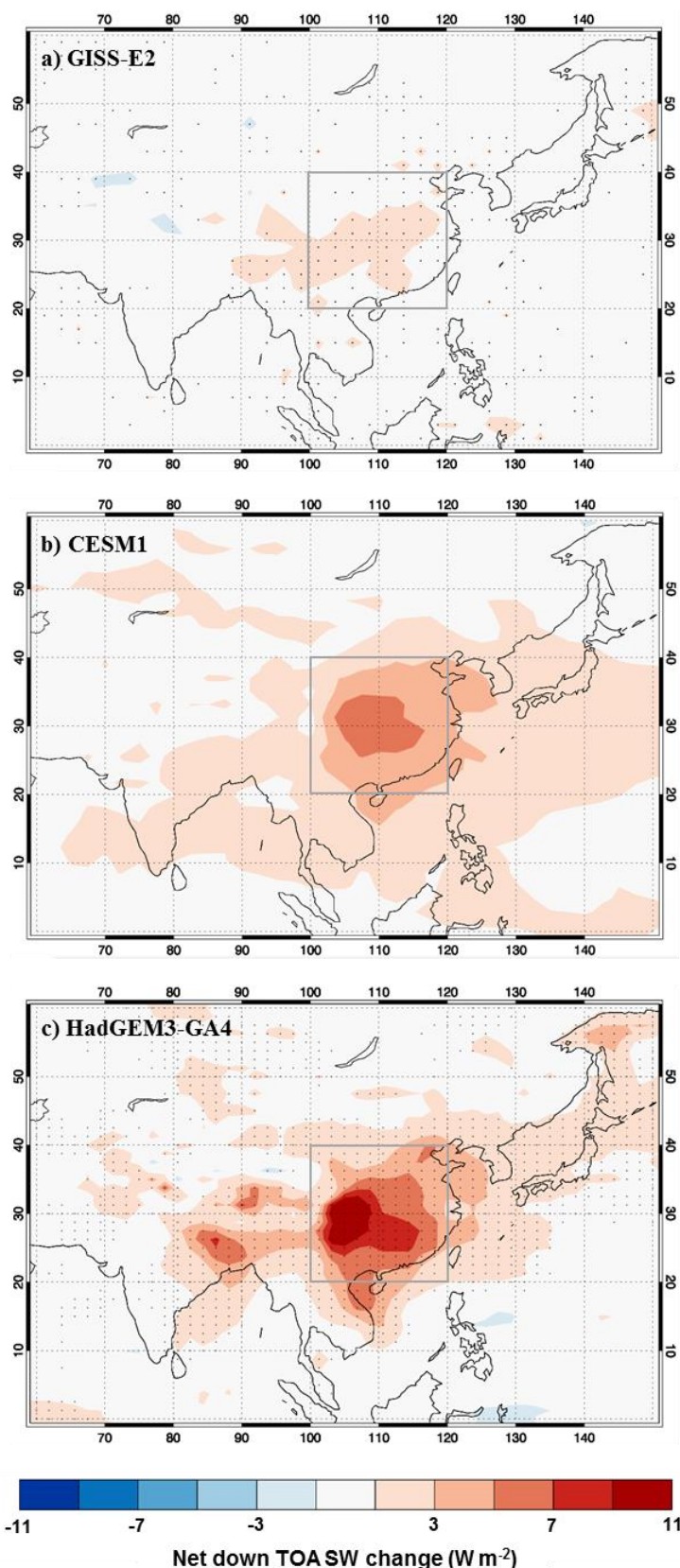

2 Figure 1: Change in net downward TOA SW flux due to removal of anthropogenic $SO_2$

3 emissions over China for a) GISS-E2, b) CESM1, and c) HadGEM3-GA4.  Differences are

calculated as the 150-year annual mean of the perturbation simulation minus the 150-year annual mean of the control simulation. Plots focus on the Asian region as changes outside this domain were found to be minimal. Stippling for GISS-E2 and HadGEM3-GA4 indicates that the change in that grid-box exceeded two standard deviations. Significance was not evaluated for CESM1 as multiple 150-year control runs were not available to assess internal variability for this model. The grey box denotes the E. China (100°E - 120°E, 20°N - 40°N) region which is used in Table 2 and throughout the discussion.

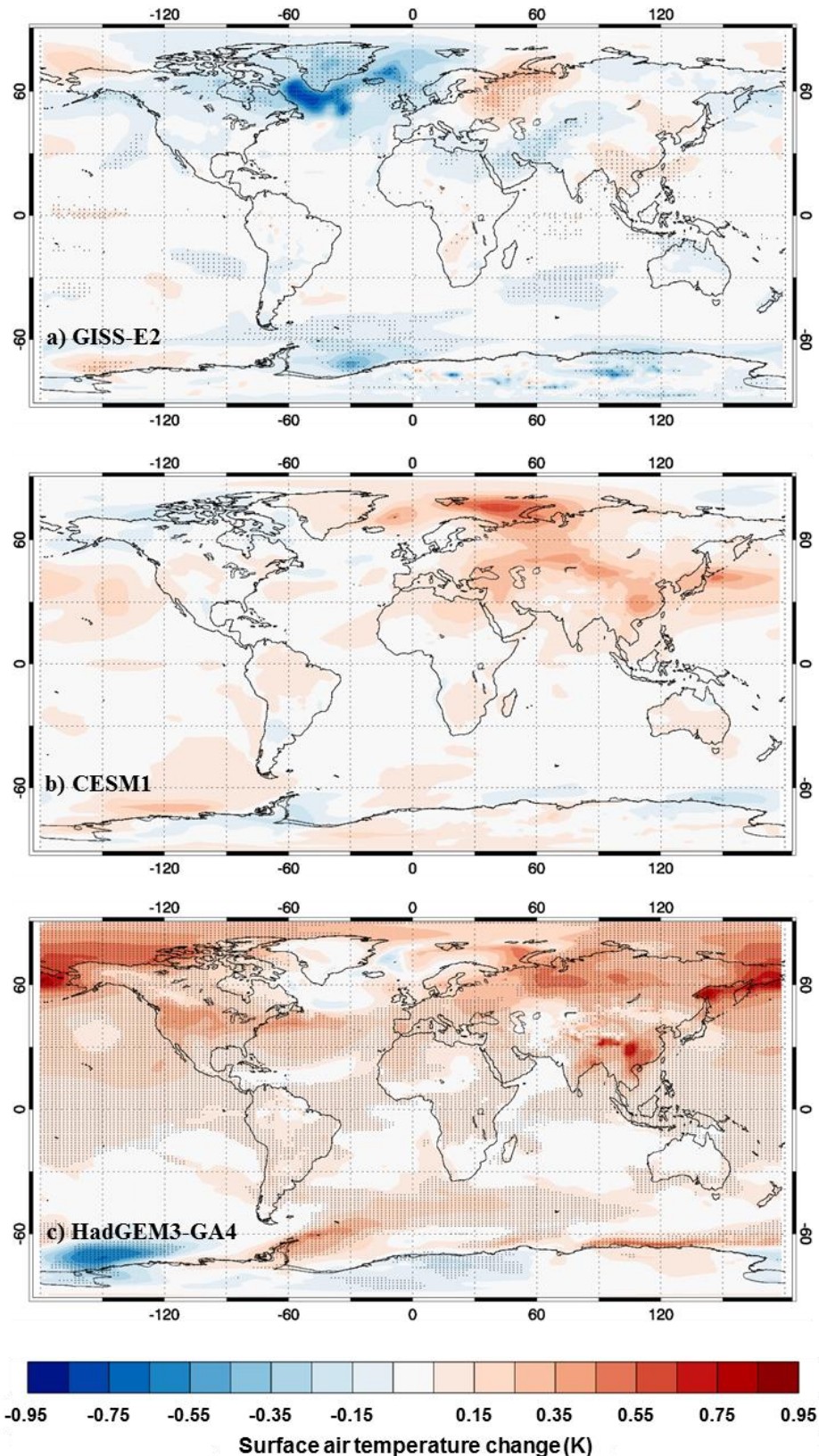

Figure 2: Global changes in surface air temperature due to removing anthropogenic SO₂ emissions from China for a) GISS-E2, b) CESM1, and c) HadGEM3.  Differences are for 150-

1    year annual means of perturbation simulation minus control simulation.  Stippling for GISS-E2

2    and HadGEM3-GA4 indicates changes exceeded two standard deviations for that grid box.

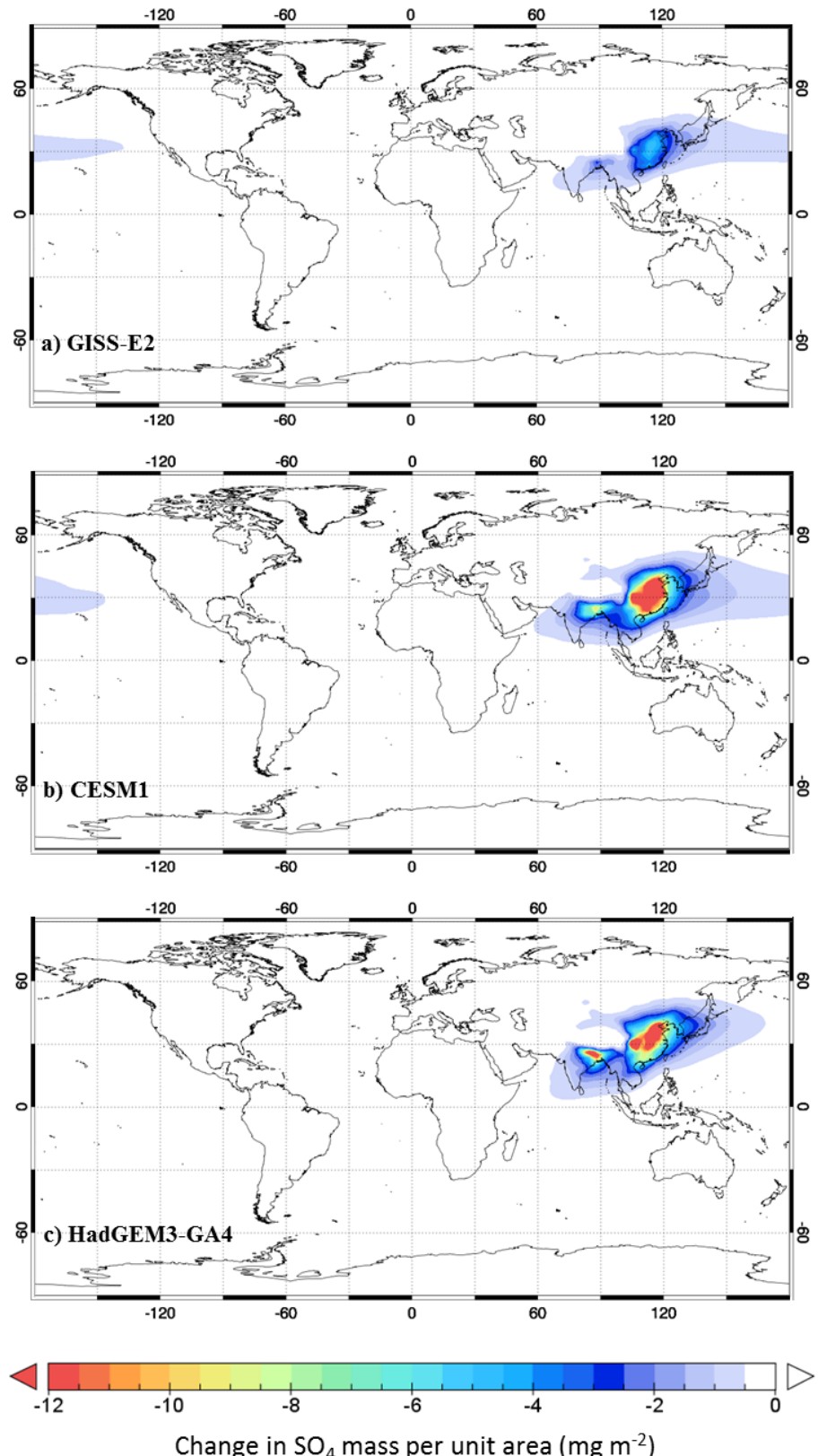

Figure 3: Global changes in column-integrated $SO_4$ burden due to removing anthropogenic $SO_2$
emissions from China, for a) GISS-E2, b) CESM1, and c) HadGEM3-GA4. Differences are
calculated as perturbation simulation minus control simulation, averaged over 150 years.

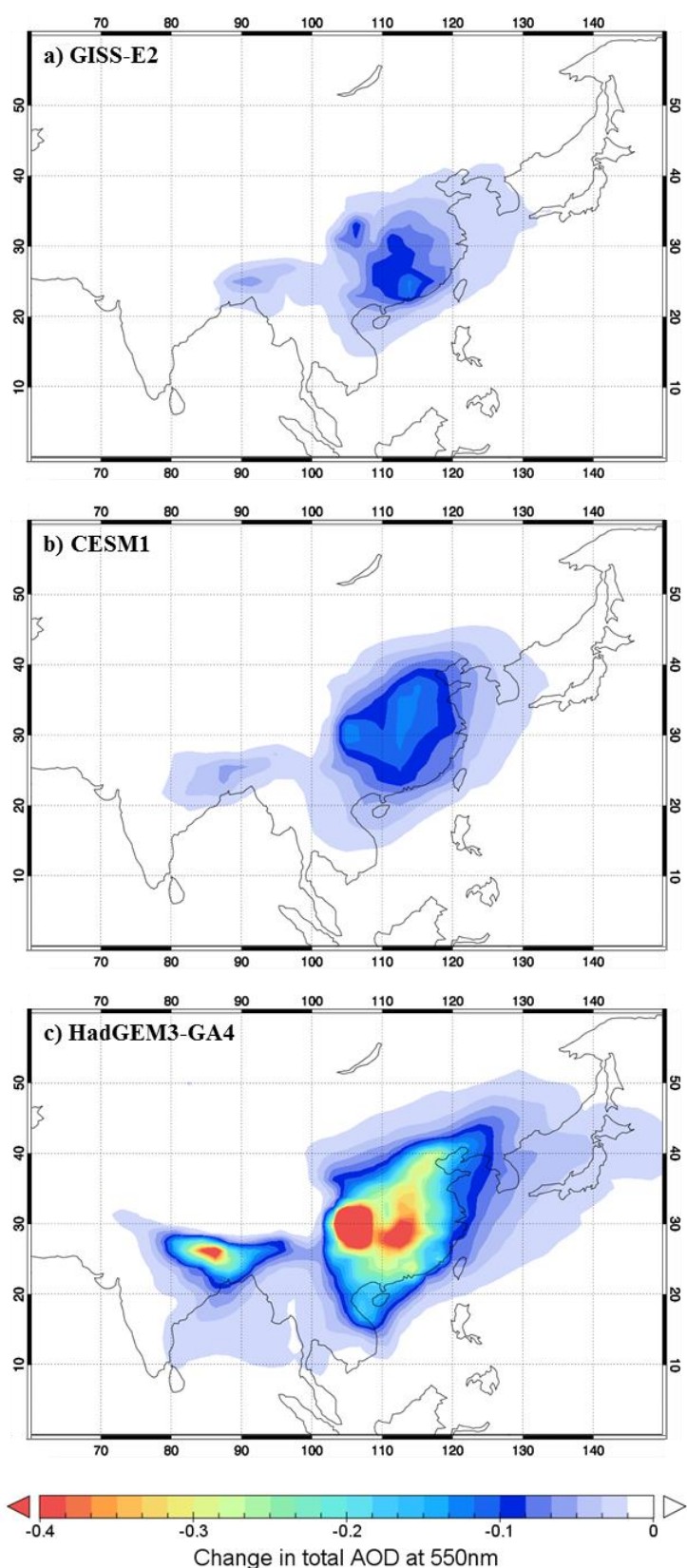

Figure 4: Change in AOD at 550nm due to removing $SO_2$ emissions from China for a) GISS-

E2, b) CESM1, and c) HadGEM3-GA4. For HadGEM3-GA4 and GISS-E2, AOD is calculated

for clear-sky conditions, whereas for CESM1 AOD is calculated for all-sky conditions, which

1    will generally result in higher values within each simulation.  Differences are calculated as

2    perturbation run minus control run, averaged over 150 years.  The plot region focuses on Asia

3    as changes outside of this domain were minimal.