# Peer review of "Regional and global temperature response to"

_Atmospheric Chemistry and Physics, 2015_

## Referee Comment (RC1) · Anonymous Referee #1 · 29 Feb 2016

Review of "Regional and global climate response to anthropogenic SO2 emissions from China in three climate models" by M. Kasoar et al.

The authors present coupled atmosphere-ocean simulations with three models to study how a removal of sulfure dioxide emission from China would impact local and global climate. Given the localized nature of aerosols and expected future reduction in sulfate aerosol, this sort of study is certainly interesting, also because the use of three models has the potential to distinguish robust and non-robust responses. As such, the study is in general suited for publication in ACP. Yet, I feel the analysis needs to go further to increase the paper's value, and a discussion about whether studies like these are indeed useful or not seems warranted at the end of the paper. What I mean by this

should become clear in my comments below.

1. The advantage of a model intercomparison study is that it allows for a clean juxtaposition of models. Yet, fundamental model diagnostics differ between the models, and I find that this very much complicates the comparison and limits the ability to draw firm conclusions beyond the statement that the models differ. I find the lack of clear-sky shortwave fluxes for CESM most striking - clearly this is a standard diagnostic, and I know that CESM has this diagnostic implemented. So why is it not available for the runs provided here? Having the clear-sky shortwave diagnostic would greatly aid the discussion of cloud effects in Sect. 4.2. Similarly, why is AOD diagnosed differently across the models, which seems to inhibit firm conclusions about AOD differences and aerosol radiative efficiencies. And finally, why is there no measure of internal variability available for CESM? I understand that this has to do with the lack of ensemble control runs (available for HadGem) or one long control run (as for GISS), but why have such runs not been performed. Aren't the authors in charge of the simulations presented here? I think the paper could be much stronger if the above limitations were addressed and the model setup and experiments were designed such as to eliminate them.

2. As a result of the above I am wondering what I am supposed to take away from the current paper, apart from the statement that there is large model uncertainty. The authors attempt to trace the uncertainty to different sources, including aerosol chemistry (Sect. 4.1), cloud-radiative effects and aerosol-cloud interactions (Sect. 4.2), aerosol-radiative interactions (Sect. 4.3) and climate sensitivity (Sect. 4.4). None of these seems to be the sole smoking gun, though. While I appreciate that there maybe is no single factor that explains most of the uncertainty, what kind of experiments would be needed to better understand the individual contributions of the above four factors? I think a discussion of this question is needed in the conclusion section.

3. Pattern of global temperature response: I am wondering to what extent the temperature patterns between the three models in Fig. 2 are more similar than acknowledged by the authors. What I mean is that GISS, while having no global-mean response,

seems to show cooling in northern hemisphere regions in which CESM and Hadgem show relatively less warming (e.g., over the North Atlantic and Iran). Maybe the temperature patterns between the models look similar when the global-mean temperature change is removed? That would be interesting and point to robustness in the remote dynamical response.

4. Reflecting on point 1, why is AOD diagnosed differently across the models? What is the motivation for this, and how to differences in the AOD diagnostics affect the results?

5. At the end of section 4.1.1, I think a statement similar to the one on page 21, lines 23-25 would be helpful to wrap up this fairly complicated subsection, which simply seems to say that comparison to observations of current AOD doesn't help to constrain the model response.

6. Sect 4.2, lines 19, "what we would expect from a simple amplification of the radiative response due to indirect effects": Clear-sky shortwave changes will always be larger than all-sky shortwave changes because clouds mask some of the aerosol. So how can a comparison between clear-sky and all-sky changes inform about aerosol-cloud interactions (i.e., indirect effects)?

7. Sect. 4.4: The idea to use global climate sensitivities derived for a uniform forcing to explain the local response to a highly localized forcings seems flawed to me to begin with, and indeed the authors find that global climate sensitivity does not help to understand the model differences. I suggest to condense this section into one or two sentences in the conclusion section.

8. Instead, I would like to encourage the authors to expand their analysis of the changes in shortwave fluxes. The diagnostic approximate shortwave model of Donohoe and Battisti, J. Climate 2011 (Atmospheric and Surface Contributions to Planetary Albedo) would be a very valuable tool to understand the contribution of atmospheric and surface reflectivity to the changes in surface flux. One can further use the model for clear-sky and all-sky fluxes separately in order to distinguish aerosol effects (from

the clear-sky use of the model) from cloud effects (when all-sky fluxes are used). I believe such an analysis has the potential to give much more insight and to grealy improve the paper.

Minor comments:

1. Information about the shortwave radiative transfer schemes is missing in the model descriptions.

2. page 8, line 1: the East China box should be drawn in one of the figures for easier reference.

3. caption figure 1: focuses –> focus

---

## Referee Comment (RC2) · Anonymous Referee #2 · 7 Mar 2016

This manuscript is a very valuable contribution to the timely research topic of the local and remote climate impacts of regional anthropogenic aerosol emission changes. The authors have done a thorough job in analyzing the causes behind the different temperature responses to an identical aerosol emission perturbation in three climate models. The results provide important new knowledge to guide further research, as well as highlight the dangers of using single models or simple proxy measures (such as precursor emissions) to estimate climate impacts. I recommend the manuscript to be published after the following minor comments have been addressed.

1) Only temperature (and no other climate) responses are addressed, and this should be reflected in also in the title.

2) The descriptions of the three models in section 2.1 should be harmonized. It is especially important to provide the readers with a detailed enough summary of the aerosol and sulfur cycle treatments in each model – currently quite little is told about CESM1 and GISS-E2 aerosol/sulphur. The treatment of aerosol-cloud interactions within each model should also be briefly summarized.

3) P5L12: What does 'mass based' scheme mean in this context when modes and bins are also treated? P5: Is aerosol microphysics (condensation, coagulation, etc.) treated in CLASSIC?

4) P6L9-10: Does this mean that chemistry is solved online? The formulation here seems overly complicated.

5) P7L1: 'aerosol-coating of dust': Dust is an aerosol particle itself; do you mean (secondary) coating of dust?

6) How different are the control climates between the different models? Would you expect this to impact your results?

7) P7L27: Are the runs restarted from an earlier simulation? 50-year spin-up by itself doesn't seem sufficient for a coupled model.

8) P10: Both HadGEM and CESM1 simulate H2O2 and O3 oxidation pathways in the aqueous phase, so including both pathways cannot be an explanation to fast conversion to SO4 in HadGEM. This should be explicitly stated.

9) P16L6-8: Do you refer to sulphate aerosol above cloud top here? Simulated cloud distributions can have large impacts also in other ways, e.g. the background aerosol amount (clean/polluted) has large impacts on indirect effects, which start to saturate at high aerosol concentrations.

10) P16L20-21: Can you speculate which dynamical processes cause the increase in cloudiness when sulphate is removed? Based on 2nd indirect effect one would assume decreased cloudiness.

---

## Referee Comment (RC3) · Anonymous Referee #3 · 13 Mar 2016

The authors present a three-model comparison of the consequences of removing anthropogenic SO2 emissions from China. They compare the radiative and climate responses, and use ground and satellite observations to try to evaluate model performance. The study is an interesting and relevant addition to the literature on the diversity of climate model responses to comparable perturbations. While some aspects of the analysis, diagnostics and presentation could have been clearer, the main results of the study are still clear: For an aerosol perturbation that is weak but realistic (i.e. not scaled, as is done in most multi-model intercomparisons), the diversity in model response is very large indeed. The paper should be published in ACP after some additions and clarifications have been made.

[Figure]

Major comments:

- While the perturbation applied is well specified, the model output and diagnostics retrieved seems to vary a lot. I realize it's hard to do anything about this once the simulations are done, but for later studies I would encourage the authors to use a wider output protocol. E.g. clear-sky vs all-sky fluxes should be possible to diagnose for all these climate models, and for sulphate perturbations their difference can be very instructive due to differences in treatment of the indirect effect.

- Page 8, line 23++: For a sudy such as this one, a good diagnostic of TOA RF is very useful. It can be extracted from relatively short and inexpensive fSST runs, as was done here for HadGEM3. I would encourage the authors to add this also for the two other models, and to take the results into their intercomparison discussions.

- Page 11, line 30-31: The GISS-E2 model has had some problems with its nitrate implementation, and e.g. pulled these results from AeroCom Phase II. Is this issue resolved for the simulations presented here? (I assume so, but still ask since nitrate here seems to be one of the drivers of intermodel differences.)

- Page 13, line 1-10: This section is very interesting, but briefly presented. I would suggest expanding it somewhat, perhaps adding some comparison plots? This would make the study even more useful for future model work.

- Page 15, line 12-30: This section discusses wet deposition results vs observations, and link good performance to a realistic SO4 distribution. However, isn't this also very dependent on the representation of precipitation? The China/Asia region has a lot of variability both in actual and modeled precipitation, and until it's shown that these compare to a reasonable degree I would be cautious about the above interpretation of wet deposition.

- Page 17, line 17-19: It's hard to assess if e.g. "a 3-fold larger clear-sky SW change" is significant without some indication of the internal variability. Since the results in this

paper are mostly from 150-year integrations, I would encourage the authors to add more information on the year-to-year variability (i.e. just the standard deviation of the result across the integration, or similar) throughout the manuscript.

- Table 1: The numbers listed here seem to have an unrealistically high precision (e.g. -0.034810...) Please give a reasonable number of significant digits, and also include some indication of the internal variability in each model (see previous comment).

Minor comments:

- Abstract (p2): "...and reinforces that caution must be applied when interpreting the results of single-model studies." I believe the results of this paper show that we should be cautions also in interpreting multi-model studies. They are usually just ensembles of opportunity, with little or no observational constraint beyond what is already taken into the model parametrizations. Hence their average values are not necessarily closer to reality, but instead just indicative of the present model diversity.

- Page 3, line 31-32: The Phase II AeroCom study (Myhre et al. 2013, ACP) which you cite later probably belongs in this company.

- Section 2.1: The description of HadGEM3-GA4 is very long compared to the two other models. Could the descriptions be clarified and made more uniform? Perhaps through a table of the most relevant model parameters/physical processes included?

- Page 18, line 1-2: The SO4 forcing is not very sensitive to the vertical distribution, compared e.g. to absorbing species. See e.g. Samset and Myhre, GRL 2011, doi:10.1029/2011GL049697.

---

## Author Comment (AC1) · 12 Jun 2016

**"Regional and global climate response to anthropogenic SO2 emissions from China in three climate models" by M. Kasoar et al.**

**Author response to anonymous referee #1**

The authors wish to express their sincere gratitude to the anonymous referee for their invaluable comments and appraisal of our study. They have provided plenty of thought-provoking points, and we very much appreciate the time taken to do so.

Below we detail our responses to each major and minor comment in turn. We hope that these responses will satisfactorily address all the points raised. The referee's comments are given in italics, below which we provide our responses and the details of any changes made in the manuscript in normal font.

**Comment 1:**

*"The advantage of a model intercomparison study is that it allows for a clean juxtaposition of models. Yet, fundamental model diagnostics differ between the models, and I find that this very much complicates the comparison and limits the ability to draw firm conclusions beyond the statement that the models differ. I find the lack of clear-sky shortwave fluxes for CESM most striking - clearly this is a standard diagnostic, and I know that CESM has this diagnostic implemented. So why is it not available for the runs provided here? Having the clear-sky shortwave diagnostic would greatly aid the discussion of cloud effects in Sect. 4.2. Similarly, why is AOD diagnosed differently across the models, which seems to inhibit firm conclusions about AOD differences and aerosol radiative efficiencies. And finally, why is there no measure of internal variability available for CESM? I understand that this has to do with the lack of ensemble control runs (available for HadGem) or one long control run (as for GISS), but why have such runs not been performed. Aren't the authors in charge of the simulations presented here? I think the paper could be much stronger if the above limitations were addressed and the model setup and experiments were designed such as to eliminate them."*

We acknowledge that with respect to some variables an ideal comparison could not be made, and the conclusions we could draw are more limited as a result, because of inconsistencies in which standard diagnostics were saved from these simulations. With regard to the most notable deficiency identified here though, we have now performed extended simulations with CESM in order to output the clear-sky shortwave fluxes for a 30-year period, and have therefore been able to substantially expand on Section 4.2 as desired.

With regard to the discrepancies in the manner AOD is diagnosed across the models, this was not the authors' choice – unfortunately clear-sky AOD was not available from the present CESM configuration, and likewise all-sky AOD is not available from the present HadGEM configuration. We certainly agree that it would have been useful to have consistent diagnostics from CESM, but we include this model in the paper because the available diagnostics nonetheless provide an interesting additional angle, although we believe the results would have been valuable even based just on the two extreme cases of HadGEM and GISS.

Performing a very long, or an ensemble of control runs with CESM would require considerable additional time. We feel that the advantage of being able to include an additional state-of-the-art

model outweighs the disadvantage of these lengthy additional simulations not yet being available. We have demonstrated a very large uncertainty in the climate model response to $SO_2$ emissions using three models. This is important to publish given the number of single model studies that have appeared recently in the literature and that have not always considered structural uncertainties in these papers. While performing additional simulations or implementing new diagnostics would certainly allow deeper investigation of the model differences, we maintain that our analysis in this paper robustly backs up the points we make in the conclusions, and that it is important to make this paper available to the community now rather than delay it.

Changes made:

1) Added CESM1 changes in clear-sky versus all-sky SW flux to Supplementary Figure S10

2) Removed sentence in Section 4.2 saying that similar comparison could not be made with CESM, and added three new paragraphs:

"The picture is different again for CESM1. Comparing the clear-sky and all-sky TOA SW flux changes for this model (Supplementary Figs. S10c,d), we find that regionally, the clear-sky changes are much smaller than the all-sky flux changes – in fact, over China the clear-sky SW flux changes in CESM1 are considerably smaller in magnitude than the clear-sky flux changes of GISS-E2 (comparing Supplementary Figs. S10a,c). Averaged over the E. China region, the clear-sky flux change in CESM1 is only 2.2 Wm$^{-2}$, compared with the 4.1 Wm$^{-2}$ clear-sky change in GISS-E2 (Table 2). However, whereas in GISS-E2 the all-sky SW flux change (0.9 Wm$^{-2}$) was then more than four times smaller than this clear-sky flux change, in CESM1 the all-sky SW flux change is instead almost two times larger than the clear-sky flux change: 4.2 Wm$^{-2}$ regionally averaged.

This is partly again due to cloud changes – in this case CESM1 has predominantly reductions in cloud amount over E. China (Supplementary Fig. S11b), which will have the effect of increasing the all-sky radiative flux change relative to the clear-sky changes. However, as with HadGEM3-GA4, these regional cloud reductions in CESM1 do not match up spatially with the maximum changes in all-sky SW flux seen in Fig. 1b and Supplementary Fig. S10d. Instead, the maximum changes in the all-sky SW flux change match closely the clear-sky SW flux changes (Supplementary Fig. S10c), which in turn correspond very well with the reduction in AOD (Fig. 4b). Both all-sky and clear-sky SW flux changes are maximum around where the AOD reduction is maximum, and in this location the all-sky flux change is still substantially greater than the clear-sky change. This implies that in CESM1 a large aerosol indirect effect, and/or effect of clouds increasing aerosol particle size through hygroscopic growth, overall amplifies the radiative effect of aerosols considerably in cloudy conditions, resulting in the much greater regional change in all-sky flux when aerosol is removed.

Between these three models, then, the way that clouds modify the radiative balance is a major source of diversity over the E. China region in the response to removing SO2 emissions from China. In GISS-E2, the inclusion of clouds greatly reduces the radiative effect of a change in sulfate aerosol. In HadGEM3-GA4, the effect of including clouds is small, and does not change the clear-sky forcing substantially. Finally in CESM1, including clouds considerably amplifies an otherwise weak clear-sky radiative flux change."

3) Removed fourth paragraph of Section 4.3, comparing CESM radiative efficiency using the all-sky flux, and replaced with new paragraph using clear-sky flux, consistent with HadGEM3 and GISS:

"CESM1 seems to sit in the middle of the range, with a regional radiative efficiency of -28.4 W m$^{-2}$ per unit AOD change (Table 2) – though again with the caveat that for CESM1, the AOD is an all-sky quantity, whereas the HadGEM3-GA4 and GISS-E2 values were calculated using clear-sky AOD. GISS-E2 provided both clear-sky and all-sky AOD diagnostics, and using instead the all-sky AOD change from GISS-E2 gives a smaller value of -22.4 W m$^{-2}$ per unit AOD, which suggests that when compared like-for-like, CESM1 (with -28.4 W m$^{-2}$) may in fact have the greater regional radiative efficiency.   More directly comparable between all three models is the regional flux change normalised by regional change in sulfate burden, which for CESM1 is -55.4 W m$^{-2}$ Tg$^{-1}$.  This is considerably lower than either HadGEM3-GA4 or GISS-E2, and indicates that the despite having at least average radiative efficiency per unit AOD, the very small translation of sulfate burden to AOD in CESM1 is a dominant factor which prevents this model from simulating a larger SW flux change and climate response than it already does.  As noted in the previous Section though, this small clear-sky flux change per unit sulfate change is compensated by a large indirect effect as well as favourable regional cloud changes, meaning that the all-sky flux change per unit AOD is by far the largest is CESM1 (Table 2), and the all-sky flux change per sulfate burden change is then average in CESM1 (not shown, but readily calculated from Table 2).  Similarly, the exceptional reduction in aerosol radiative effects due to clouds in GISS-E2 means that its all-sky flux change per unit AOD is almost exactly the same as that of HadGEM3-GA4 (Table 2), despite the clear-sky regional radiative efficiency being so much larger."

4) Added clear-sky flux changes for all three models to Table 2 (formerly Table 1)

**Comment 2:**

"*As a result of the above I am wondering what I am supposed to take away from the current paper, apart from the statement that there is large model uncertainty. The authors attempt to trace the uncertainty to different sources, including aerosol chemistry (Sect. 4.1), cloud-radiative effects and aerosol-cloud interactions (Sect. 4.2), aerosol-radiative interactions (Sect. 4.3) and climate sensitivity (Sect. 4.4). None of these seems to be the sole smoking gun, though. While I appreciate that there maybe is no single factor that explains most of the uncertainty, what kind of experiments would be needed to better understand the individual contributions of the above four factors? I think a discussion of this question is needed in the conclusion section.*"

Indeed, we believe we show that there is no single smoking gun, but several different factors which contribute to the uncertainty, which are all important.  We reiterate again that this is the first time such a comparison has been made between three different models forced with the same regional emissions change, and so even the statement that the models differ considerably in their responses, and for a complicated mixture of reasons, is we believe an interesting finding from the available data.  If the situation is that the response is very diverse because of several different reasons, this is important to document, even if it is not a simple conclusion.  However, we have clarified the conclusions to better highlight what appear to be the largest sources of disparity.  We agree also that some additional discussion in the conclusions of how further experiments could help elucidate this problem is worthwhile, and this has also been added.

Changes made:

1) Changes to third paragraph of the Conclusion as shown by markup below:

"Specifically, we find that CESM1 simulates the largest reduction in sulfate burden both globally and locally.  HadGEM3-GA4 has the smallest reduction in sulfate burden globally and the second largest reduction regionally, yet it produces by far the largest reduction in AOD both globally and regionally over E. China.   Though  GISS-E2 and CESM1 both simulate much smaller changes in AOD than HadGEM3-GA4, still the SW flux changes and temperature responses produced are very different between these two models.  An inferred larger aerosol-cloud interaction means that CESM1 simulates a particularly large change in all-sky SW flux relative to its fairly small AOD change, so although having a smaller response than HadGEM3-GA4, it is still much closer to it than GISS-E2.  In GISS-E2 the clear-sky radiative forcing efficiency of sulfate is very large, but this is almost perfectly compensated for by large reductions in the direct radiative effect of sulfate when clouds are factored in.  The absolute AOD change is also much smaller than HadGEM3-GA4 in this model, and this then combines with compensating increases in  and nitrate aerosol globally to reduce the radiative response yet further, and finally a smaller global climate sensitivity than the other two models results in this being translated into a largely negligible temperature response."

2) Split second paragraph of Conclusions into two, and moved the second half ("In addition to differences in sulfate and AOD…") after the third paragraph.

3) In the paragraph after, in the sentence "However, the main conclusion is that comparison against all existing observational measures is unable to satisfactorily constrain which model response is more realistic", added:

", given that the ratios of both AOD change per sulfate burden change and SW flux change per AOD (Table 1) are found to vary so substantially between the models"

4) Added new paragraph to the Conclusions:

"There are a number of possible avenues for future work to isolate the particular processes that lead to this model diversity in more detail; for instance studies imposing the aerosol field from one model into others would remove the diversity introduced by translating emissions into aerosol concentrations, while imposing surface temperatures and meteorology from one model into others could remove the diversity introduced by different background climatologies and climate sensitivities, although this may be difficult practically in complex climate models.  A thorough assay of the range of parameter choices and formulae used in the aerosol schemes of various models could also help reveal where assumed aerosol properties diverge.  However, without stronger observational constraints on aerosol radiative forcing, it is not clear that this alone could help make models more realistic.  In particular, it seems that being able to better constrain the column-integrated sulfate burden, the AOD per sulfate burden, and the radiative forcing per AOD, would all also

be needed.  This represents a considerable observational challenge, and until it is possible, the considerable current diversity may be irreducible."

**Comment 3:**

*"Pattern of global temperature response: I am wondering to what extent the temperature patterns between the three models in Fig. 2 are more similar than acknowledged by the authors. What I mean is that GISS, while having no global-mean response, seems to show cooling in northern hemisphere regions in which CESM and Hadgem show relatively less warming (e.g., over the North Atlantic and Iran). Maybe the temperature patterns between the models look similar when the global-mean temperature change is removed? That would be interesting and point to robustness in the remote dynamical response."*

This is an interesting suggestion, and we have now taken a look at this, but unfortunately it doesn't seem to show anything different – see plots below.  Part of the problem we think is that what is seen in GISS is not really a response at all, but almost entirely noise.

[Figure]

Temperature change minus global mean (K)

**Comment 4:**

*"Reflecting on point 1, why is AOD diagnosed differently across the models? What is the motivation for this, and how to differences in the AOD diagnostics affect the results?"*

The first part of this is already addressed in the responses to Comment 1 – there was no deliberate motivation on the part of the authors, but unfortunately these are the diagnostics that were available from these model versions.  And we still believe that the comparison is valuable.  Regarding the second point here, it is consequently very difficult to know exactly how this will affect the results, however in Section 4.1 we do make comparison between the GISS-E2 all-sky AOD and CESM1 AOD, which should be more directly comparable, and we also note from the differences between the all-sky and clear-sky diagnostics in GISS-E2 that an all-sky diagnostic is likely to give larger values than the equivalent clear-sky diagnostic.

**Comment 5:**

*"At the end of section 4.1.1, I think a statement similar to the one on page 21, lines 23-25 would be helpful to wrap up this fairly complicated subsection, which simply seems to say that comparison to observations of current AOD doesn't help to constrain the model response."*

This has been added.

Changes made:

1) Added new paragraph at end of Section 4.1.1:

   " Still, overall HadGEM3-GA4 seems to compare slightly better than GISS-E2 and CESM1 regionally over E. Asia against observations of total AOD, and better than GISS-E2 regionally against surface sulfate as well as wet deposition observations, although globally and over other regions this model is not necessarily found to compare better in general.  This might hint that at least over China, HadGEM3-GA4 has more realistic sulfate optical depth, although none of these comparisons is very conclusive in that respect.  Moreover, given that none of these observational measures directly constrains the sulfate radiative forcing, there is also no guarantee that performance with respect to these variables will necessarily translate to a more realistic climate response (see also Section 4.3)."

2) For greater clarification of the statement in the conclusion, also added an additional sentence at end of first paragraph of Section 4.3:

   "As a result, whether a model simulates AOD changes correctly, for instance, may not particularly constrain the resultant forcing and eventual climate response."

**Comment 6:**

*"Sect4.2, lines19, "what we would expect from a simple amplification of the radiative response due to indirect effects": Clear-sky shortwave changes will always be larger than all-sky shortwave changes because clouds mask some of the aerosol. So how can a comparison between clear-sky and all-sky changes inform about aerosol-cloud interactions (i.e., indirect effects)?"*

We agree that the highlighted sentence needed to be removed, as it is indeed mistaken.  However we do still believe that the comparison made in the rest of this section, of the differences in the relative magnitudes of all-sky and clear-sky fluxes between the models, tells us something useful about the importance of cloud effects – although one cannot distinguish cleanly between microphysical and dynamical effects.  (Indeed, the reviewer in their first comment also noted that: "Having the clear-sky shortwave diagnostic would greatly aid the discussion of cloud effects in Sect. 4.2", and so they presumably agree that something can be concluded from making such a comparison).  In fact, the clear-sky flux changes need not necessarily be larger than the all-sky change if indirect effects are larger than direct effects, and this indeed seems to be the case for CESM, from the newly-added clear-sky diagnostics.

Changes made:

1) Removed:

   "In fact, in both models the clear-sky SW change turns out to be larger than the all-sky SW change, which is opposite to what we would expect from a simple amplification of the radiative response due to indirect effects.  In particular GISS-E2 simulates an increase in cloudiness in East China when sulfate is removed, which…"

2) Replaced with:

"…compared with the clear-sky change, the all-sky response incorporates all the contributing factors described above: the additional radiative forcing due to aerosol indirect effects, the screening of direct radiative effects due to clouds blocking radiation and providing a high albedo background, and also any dynamical changes in cloud cover.

In this case, GISS-E2 is found to simulate a small increase in cloudiness in east China due to dynamical changes when sulfate is removed (Supplementary Fig. S11a). Combined with the screening effect of clouds, this…"

**Comment 7:**

"*Sect. 4.4: The idea to use global climate sensitivities derived for a uniform forcing to explain the local response to a highly localized forcings seems flawed to me to begin with, and indeed the authors find that global climate sensitivity does not help to understand the model differences. I suggest to condense this section into one or two sentences in the conclusion section.*"

The reviewer notes that we find the use of global climate sensitivities derived from a uniform forcing to be not particularly helpful in understanding the model differences – particularly between HadGEM3-GA4 and CESM1 (although GISS-E2 does have a known low climate sensitivity, which probably does contribute to this model having the lowest response along with the other factors discussed). However, we believe section is important partly to highlight this very fact. The comparison may be flawed, but yet global climate sensitivities are still typically used – very few studies have ever tried to calculate or use regional sensitivities. In meta-reviews like the IPCC AR5, it is typically implicitly assumed that the forcing due to inhomogeneous species like aerosols can be summed up with a global mean value for the forcing. As a result we believe this section still has value to draw attention to this. We already stress in this section that the comparison is flawed and that the global climate sensitivity to a uniform forcing should not be considered as equivalent to the climate sensitivity to a localised forcing, and highlight the lack of studies that have explored this issue.

**Comment 8:**

"*Instead, I would like to encourage the authors to expand their analysis of the changes in shortwave fluxes. The diagnostic approximate shortwave model of Donohoe and Battisti, J. Climate 2011 (Atmospheric and Surface Contributions to Planetary Albedo) would be a very valuable tool to understand the contribution of atmospheric and surface reflectivity to the changes in surface flux. One can further use the model for clear-sky and all-sky fluxes separately in order to distinguish aerosol effects (from the clear-sky use of the model) from cloud effects (when all-sky fluxes are used). I believe such an analysis has the potential to give much more insight and to grealy improve the paper.*"

We appreciate the reviewer's thoughts on potential further ways to expand on our analysis. We have considered the method suggested, but ultimately feel that our analysis in this paper already robustly backs up the points we make in the conclusions. Surface reflectivity changes appear to be unimportant to the responses over the East Asian region that we analyse (instance.g. we have verified, at least in HadGEM and GISS, that the local surface albedo is almost exactly the same in

control and perturbation simulations), so in this case we do not feel that using the suggested additional model would change our analysis.

**Minor comment 1:**

"*Information about the shortwave radiative transfer schemes is missing in the model descriptions.*"

This information has been added.

Changes made:

1) Added to HadGEM3 model description:

   "The radiative transfer scheme of Edwards and Slingo (1996) is used with six spectral bands in the shortwave, and…"

2) Added to CESM1 model description:

   "Shortwave radiative transfer is based on the RRTM_SW scheme (Clough et al., 2005) with 14 spectral bands, and aerosols interact with climate through both absorption and scattering of radiation."

3) Added to GISS model description:

   "Aerosols direct effects are calculated following the Hansen et al. (1983) radiation model, with six spectral bands in the shortwave."

4) Added Edwards and Slingo (1996), Clough et al. (2005) and Hansen et al. (1983) to reference list.

**Minor comment 2:**

"*page 8, line 1: the East China box should be drawn in one of the figures for easier reference.*"

Done.

Changes made:

1) Box showing outline of E. China region added to all panels of Fig. 1.

2) Added to caption of Fig. 1:

   "The grey box denotes the East China (100°E - 120°E, 20°N - 40°N) region which is used in Table 1 and throughout the discussion."

3) Added sentence to end of second paragraph of Section 3 (where Fig. 1 is introduced):

   "For reference, Fig. 1 also shows the outline of the E. China region, which corresponds well to the region of maximum SW flux changes in all three models."

**Minor comment 3:**

*"caption figure 1: focuses –> focus"*

Corrected.

Changes made:

1) 'focuses' changed to 'focus' in Fig. 1 caption.

---

## Author Comment (AC2) · 12 Jun 2016

**"Regional and global climate response to anthropogenic SO2 emissions from China in three climate models" by M. Kasoar et al.**

**Author response to anonymous referee #3**

The authors wish to express their sincere gratitude to the anonymous referee for their invaluable comments and positive appraisal of our study. They have provided thorough and thought-provoking points and we very much appreciate the time taken to do so.

Below we detail our responses to each major and minor comment in turn. We hope that these responses will satisfactorily address all the points raised. The referee's original comment is included in italics, with our response and change to the manuscript in normal font.

**Comment 1:**

*"While the perturbation applied is well specified, the model output and diagnostics retrieved seems to vary a lot. I realize it's hard to do anything about this once the simulations are done, but for later studies I would encourage the authors to use a wider output protocol. E.g. clear-sky vs all-sky fluxes should be possible to diagnose for all these climate models, and for sulphate perturbations their difference can be very instructive due to differences in treatment of the indirect effect."*

We absolutely agree. This shortcoming was mentioned also by the first reviewer, and the lessons from this study are indeed being learnt in the discussion of potential future collaborations, which have developed following presentations of the results in this study. As described in the responses to Referee #1, we have in fact also taken the step of extending the simulations with CESM1 for a short period to diagnose the previously missing clear-sky SW flux (which we expect has lower variability than temperature, and so probably doesn't need the same 150-year averaging period), and so the discussion in Sections 4.2 and 4.3 has been updated with this new data.

**Comment 2:**

*"Page 8, line 23++: For a sudy such as this one, a good diagnostic of TOA RF is very useful. It can be extracted from relatively short and inexpensive fSST runs, as was done here for HadGEM3. I would encourage the authors to add this also for the two other models, and to take the results into their intercomparison discussions."*

We do agree with the reviewer that including additional simulations would be helpful to get a more precise measure of the radiative responses. However, we have opted already to use the available time to extend the coupled simulations with CESM in order to diagnose clear-sky fluxes as requested by the first reviewer, which we decided was a more critical deficiency. Although more thorough RF diagnostics would be nice for consistency, we do not anticipate they would qualitatively change any of our findings, and we strongly believe that our analysis with the presently available diagnostics already robustly supports the points we make in the conclusions. Given the number of single model studies that have appeared recently in the literature and that have not always considered structural uncertainties, we believe these conclusions are already of sufficient importance and urgency to merit publishing this paper now, rather than incur the further delay and additional costs of additional simulations.

**Comment 3:**

"*Page 11, line 30-31:  The GISS-E2 model has had some problems with its nitrate implementation, and e.g. pulled these results from AeroCom Phase II. Is this issue resolved for the simulations presented here?  (I assume so, but still ask since nitrate here seems to be one of the drivers of intermodel differences.)*"

The GISS-E2 configuration used here is the AR5 version, meaning that it does still suffer from the issue of too high a nitrate burden, and probably an overly strong nitrate response as a result.  This was, in fact, one of the first things we considered as a possible cause of the discrepancy between GISS and the other models.  However, as we discuss in the paper, although there is some partial compensation by increases in nitrate, it turns out to still be a fairly minor factor in the inter-model differences in this study.

**Comment 4:**

"*Page 13, line 1-10:  This section is very interesting, but briefly presented.  I would suggest expanding it somewhat, perhaps adding some comparison plots? This would make the study even more useful for future model work.*"

P13, L1-10 discusses the comparison against AERONET, for which there is already a comparison plot in the supplement and we are not sure that there is much scope to expand on it.  However, we think the reviewer may have meant Page 14, where we discuss the fractional change in AOD, which turns out to be much larger in HadGEM3 than in GISS-E2 or CESM1.  In this case then yes, we do agree that this was rather interesting and could merit some more detail.  We have therefore expanded the discussion and added two extra Supplementary Figures here showing firstly how the sulfate fraction of total AOD varies considerably between HadGEM3-GA4 and GISS-E2, and then also comparing the non-sulfate AOD to show that this is in fact similar in these two models, and so the discrepancy in the fraction of total AOD removed is primarily due to disagreeing on the sulfate optical depth only.

Changes made:

1) At the end of the third paragraph of Section 4.1.1, added:

   "This is illustrated further for the two extreme cases, HadGEM3-GA4 and GISS-E2, in Supplementary Fig. S3, which shows that the fraction of climatological AOD made up by sulfate is consistently higher across the east Asian region in HadGEM3-GA4 than in GISS-E2. However, the total non-sulfate AOD is fairly similar across the region in these two models (Supplementary Fig. S4), indicating that the stark difference in the fractional contribution of sulfate comes primarily from HadGEM3-GA4 simulating much greater sulfate AOD alone. Given that regionally GISS-E2 appeared to underestimate total AOD, this would then suggest that either the higher sulfate AOD in HadGEM3-GA4 is more realistic, or else both models underestimate the non-sulfate AOD."

2) Added new Supplementary Figure (S3), showing fraction of total AOD made up by sulfate in GISS-E2 and HadGEM3-GA4.

3) Added new Supplementary Figure (S4) showing total non-sulfate AOD (i.e. total AOD minus sulfate AOD) in GISS-E2 and HadGEM3-GA4.

4) Renumbered other Supplementary Figures accordingly.

**Comment 5:**

"*Page 15, line 12-30:  This section discusses wet deposition results vs observations, and link good performance to a realistic SO4 distribution. However, isn't this also very dependent on the representation of precipitation?  The China/Asia region has a lot of variability both in actual and modeled precipitation, and until it's shown that these compare to a reasonable degree I would be cautious about the above interpretation of wet deposition.*"

A valid point.  We have added a caveat to this part of our discussion by noting that precipitation will influence the amount of local wet deposition, and so it is difficult to draw definite conclusions from this comparison.  (Although, because wet deposition is the primary sink of sulfate aerosol, to some extent regionally it must balance the source of aerosol regardless of the precipitation, and so a large underestimate in the amount of wet deposition could be indicative of too low production of sulfate aerosol).  At any rate, we do not rely on this single measure to determine which model is more accurate, but note that it appears consistent with the other observations that we compare with in suggesting that GISS-E2 likely simulates too little sulfate in the region.

Changes made:

1) Added sentence to end of wet deposition paragraph:

   " This overall picture seems consistent with that of the other observational measures looked at here, although it should be noted that wet deposition rates are dependent not just on the column sulfate burden but also on the amount and distribution of precipitation however, and so biases in wet deposition could also be due to incorrect precipitation distribution rather than sulfate."

**Comment 6:**

"*Page 17, line 17-19: It's hard to assess if e.g. "a 3-fold larger clear-sky SW change" is significant without some indication of the internal variability. Since the results in this paper are mostly from 150-year integrations, I would encourage the authors to add more information on the year-to-year variability (i.e. just the standard deviation of the result across the integration, or similar) throughout the manuscript.*"

We agree with the reviewer that some desirable detail on the significance of the results was either omitted or hard to find, which we have tried to rectify.  In our SW and surface temperature plots for GISS-E2 and HadGEM3 we did already include a measure of significance by stippling the plots and stated in the text which temperature responses were significant, but we have now extended that by including ± 2σ uncertainty values in the Table of global and regional responses for all variables that there were sufficient data to calculate it for.  This includes the clear-sky SW changes in HadGEM3 and GISS, for which the discrepancy is seen to be extremely significant (around 23 standard deviations).  Extending this to all variables in the Table is complicated by the fact that the very long

control simulation used to assess variability in GISS-E2 only output basic climate diagnostics and not more detailed aerosol-related diagnostics, and we have no equivalent long or ensemble control simulation at all for CESM.  The SW changes and final temperature response are ultimately what we are interested in most though, so we do not think that this is too restrictive (and one can generally use the value given for HadGEM3 to get at least an order-of-magnitude estimate of the likely uncertainty where a value isn't available for the other models).  We deliberately avoided estimating the significance of other variables from the year-to-year variability in these simulations though, because we do not think this necessarily leads to an accurate measure of the long-term 150-year variability which is the relevant quantity here, and on which we base our uncertainty analysis.

Changes made:

1) Added ± 2σ uncertainty values to the Ch0-Con differences in Table 2 (formerly Table 1), for all variables for which long/multiple control runs data were available (all of HadGEM3 + temperature and radiative fluxes for GISS).  Added statement to Table 2 caption:

   "For models and variables where data was available, error ranges are quoted for the Ch0-Con values and indicate ± 2 standard deviations, evaluated in HadGEM3-GA4 from an ensemble of six 150-year control runs with perturbed initial conditions, and in GISS-E2 from twelve 150-year segments of a long pre-industrial control run.  Values quoted without error ranges indicate that uncertainty was not evaluated."

**Comment 7:**

"*Table 1: The numbers listed here seem to have an unrealistically high precision (e.g. -0.034810…) Please give a reasonable number of significant digits, and also include some indication of the internal variability in each model (see previous comment).*"

We agree that the precision that the numbers are quoted to is implausibly high – this is an oversight that appears to have crept in from an old version of the table, and the numbers should have been truncated to fewer significant figures in the submitted version.  This has now been corrected.  See our response to the previous comment for discussion of internal variability and estimating significance – we have added error values into Table 1 for the variables and models for which these we had these figures.

Changes made:

1) Values in Table 2 truncated so that Ch0-Con values are at most 3 significant figures.  Values for individual simulations have been truncated to at most the same number of decimal places as the Ch0-Con anomalies for that variable.

2) Added significance estimates to Table 2 as detailed in response to Comment 6

**Minor comment 1:**

"*Abstract (p2): "…and reinforces that caution must be applied when interpreting the results of single-model studies." I believe the results of this paper show that we should be cautions also in interpreting multi-model studies. They are usually just ensembles of opportunity, with little or no observational*

*constraint beyond what is already taken into the model parametrizations. Hence their average values are not necessarily closer to reality, but instead just indicative of the present model diversity.*"

We have modified both the abstract and conclusion so as to not limit our statement to single-model studies.

Changes made:

1) In the abstract, changed 'single-model studies' to 'modelling studies'

2) Changed the corresponding line in the second last paragraph in the conclusion ("…and imply that care must be taken not to over-interpret the results of studies performed with single models") to:

   "…and imply that care must be taken not to over-interpret studies of aerosol-climate interaction if the robustness of results across diverse models cannot be demonstrated"

**Minor comment 2:**

"*Page 3, line 31-32: The Phase II AeroCom study (Myhre et al. 2013, ACP) which you cite later probably belongs in this company.*"

We agree, and have added a reference to this paper in that section as well.

Changes made:

1) Added 'Myhre et al., 2013' to bracketed list of HTAP and AeroCom references.

**Minor comment 3:**

"*Section 2.1: The description of HadGEM3-GA4 is very long compared to the two other models. Could the descriptions be clarified and made more uniform? Perhaps through a table of the most relevant model parameters/physical processes included?*"

Agreed – this is something that has been mentioned by another reviewer as well, though the other reviewer favoured more detail for CESM1 and GISS-E2 rather than less for HadGEM3-GA4. We have slightly cut down superfluous details in the HadGEM3 description while adding several additional details to the other model descriptions and slightly re-ordering them to make the descriptions more uniformly structured. As recommended, we have also added a new table which includes key references and features of the three models for easy reference.

Changes made:

1) Numerous changes to model descriptions which are detailed in responses to Referee #1 Minor Comment 1 and Referee #2 Minor Comments 2, 3, 4, and 5, which harmonise the model descriptions.

2) In the first paragraph of Section 2, added:

   "The models are briefly described below, and the key references and features are also

summarised in Table 1."

3) Added a new table (Table 1; previous Table 1 is now Table 2) with key model details. Updated all previous instances of "Table 1" in the text to "Table 2", and updated the caption of the existing table to Table 2. Added caption to new table:

"Table 1:  Key references and features of the three models and their aerosol schemes used in this study"

**Minor comment 4:**

"*Page 18, line 1-2: The SO4 forcing is not very sensitive to the vertical distribution, compared e.g. to absorbing species. See e.g. Samset and Myhre, GRL 2011, doi:10.1029/2011GL049697.*"

This is very true.  We have removed that speculation, and found a different (partial) explanation:

Changes made:

1) Removed "For instance, the forcing per unit AOD will be influenced by the vertical distribution of the aerosol (Myhre et al., 2013a), which could vary between models in different parts of the world."

2) Replaced with:

"The sulfate efficiencies in Myhre et al. (2013) are calculated relative to all-sky direct radiative effect, and so local differences in vertical profiles and cloud screening may therefore change the relationship – however they also evaluated clear-sky forcing normalised by AOD for all aerosol species combined, and again found HadGEM2 to be higher than GISS ModelE."

3) Additionally, at end of this section, added text indicated in the mark-up below:

"However, the study also found that, globally, the atmospheric component of HadGEM2 had a slightly larger forcing efficiency to CAM5.1 both for sulfate (all-sky) and all aerosols (clear-sky), unlike the somewhat smaller regional efficiency found here for HadGEM3-GA4 compared with CESM1.  Given that our regional values from GISS-E2 and HadGEM3-GA4 also seem to conflict qualitatively with the global values from the AeroCom study, this would suggest that either the global comparison is not relevant on regional scales, or else the radiative efficiency is very sensitive to changes in model configuration and version.

---

## Author Comment (AC3) · 12 Jun 2016

**"Regional and global climate response to anthropogenic SO2 emissions from China in three climate models" by M. Kasoar et al.**

**Author response to anonymous referee #2**

The authors are extremely grateful to the reviewer for their extremely helpful and positive comments. We very much appreciate the time taken to do provide these comments, which have helped highlight some areas of the paper where we were unclear or not precise enough.

Below we detail our responses to each minor comment in turn. We hope that these responses will satisfactorily address all the points raised. The referee's comments are included in italics, with our response to them and relevant changes to the manuscript in normal font.

**Minor comment 1:**

"*Only temperature (and no other climate) responses are addressed, and this should be reflected in also in the title.*"

Modified.

Changes made:

1) 'Climate' changed to 'temperature' in the title

**Minor comment 2:**

"*The descriptions of the three models in section 2.1 should be harmonized. It is especially important to provide the readers with a detailed enough summary of the aerosol and sulfur cycle treatments in each model – currently quite little is told about CESM1 and GISS-E2 aerosol/sulphur. The treatment of aerosol-cloud interactions within each model should also be briefly summarized.*"

We have attempted to harmonise the descriptions of the models through providing some additional details on CESM1 and GISS-E2, while slightly cutting down unnecessary text in the HadGEM3 description (we note that another reviewer actually thought our description of HadGEM3 was already too long, and so providing the right level of detail without hurting the flow and main message of the paper is a difficult balance). We have also, at the suggestion of the third reviewer, collated key details of the models into a table for easier reference.

Changes made:

1) In HadGEM3-GA4 description, removed:

   "..., dynamically resolving the stratosphere"

   "..., which includes 4 soil layers and 5 plant functions types. Although in principle this can be run in a fully interactive 'Earth System' mode with dynamic vegetation and a carbon cycle,..."

   "More detailed description and evaluation of the atmosphere and land surface schemes can be found in Walters et al. (2014)."

"Critical to our study is the representation of aerosols; we…"

"…, which is described and evaluated in…"

"The remaining aerosol species are emitted directly in the particulate phase, and…"

"…can then undergo advection, wet and dry deposition, and…"

2) In HadGEM3-GA4 description, inserted:

"(Walters et al., 2014)" in first and second sentences.

"HadGEM3-GA4 can be run with a choice of two aerosol schemes of differing complexity – CLASSIC (Bellouin et al., 2011), and GLOMAP (Mann et al., 2010).  Here we use the simpler CLASSIC scheme, which is less computationally expensive, and is also the aerosol scheme that was used for CMIP5 simulations with the predecessor of this model (HadGEM2). CLASSIC is a mass-based scheme, meaning that only aerosol mass (and not particle number) is tracked, and therefore all aerosol species are assumed to be externally mixed."

"…mass…" in the sentence: "Cloud droplet number concentration and effective radius are determined from the mass concentration of these aerosols…"

plus minor connecting words so that sentences still read correctly after the phrases removed above.

3) In CESM1 description, removed:

"…modal aerosol scheme…"

"…from anthropogenic and natural…"

4) In CESM1 description, added:

"CAM5-Chem uses the MAM3 modal aerosol scheme (Liu et al., 2012), which is the same as used for the CESM1 submission to CMIP5.  Both aerosol mass and particle number are prognostic, and the scheme simulates sulfate, black carbon, primary organic matter, secondary organic aerosol, dust, and sea salt aerosol species as an internal mixture in Aitken, accumulation, and coarse modes."

"The model includes emissions of natural and anthropogenic SO2 and natural DMS as sulfate precursors, and…"

"Aerosols-cloud interactions allow for the effect of aerosols on both cloud droplet number and mass concentrations (Tilmes et al, 2015)."

5) In GISS-E2 description, split second paragraph in to two and moved "nitrate, elemental and organic carbon along with secondary organic aerosols and natural sea-salt and mineral dust"

from the last paragraph to the new third paragraph.

6) In GISS-E2 description, replaced "$SO_2$ from anthropogenic and natural sources…" with "$SO_2$ from these sources…"

7) In GISS-E2 description, added:

"GISS-E2 has a choice of three aerosol schemes of varying complexity – OMA (Koch et al., 2011; 2006), MATRIX (Bauer et al, 2008), and TOMAS (Lee and Adams, 2012). Following the GISS-E2 CMIP5 configuration, we use here simpler mass-based OMA scheme, which includes sulfate, …"

"Aerosols are parameterised as an external mixture of dry and dissolved aerosol, with particle size parameterised as a function of relative humidity (Schmidt et al., 2006)."

"includes natural emissions of DMS, and natural and anthropogenic emissions of SO2."

"…, such that cloud droplet number concentration and autoconversion rate depend on the local concentration of aerosol."

8) Added a new table (Table 1; previous Table 1 is now Table 2) with key model details, as described in response to Referee #3 Minor Comment 3.

**Minor comment 3:**

"*P5L12: What does 'mass based' scheme mean in this context when modes and bins are also treated? P5: Is aerosol microphysics (condensation, coagulation, etc.) treated in CLASSIC?*"

We mean that only the mass concentration of each aerosol species (as opposed to number concentration) is tracked within each of the Aitken, accumulation, and dissolved modes. We have clarified this part of the description. Because only the mass of aerosol within each mode is tracked, microphysics is parameterised to allow transfer of mass between the different modes, based on the mass concentrations of each mode.

Changes made (also included in response to Minor comment 2 above):

1) Added "CLASSIC is a mass-based scheme, meaning that only aerosol mass (and not particle number) is tracked, and therefore all aerosol species are assumed to be externally mixed" in the description of CLASSIC

**Minor comment 4:**

"*P6L9-10: Does this mean that chemistry is solved online? The formulation here seems overly complicated.*"

Yes – this has been clarified.

Changes made:

1) In CESM1 description, added "…online…" in "…we use an online representation of tropospheric and stratospheric chemistry…"

**Minor comment 5:**

*"P7L1: 'aerosol-coating of dust': Dust is an aerosol particle itself; do you mean (secondary) coating of dust?"*

Yes – have amended to clarify.

Changes made:

1) Changed "aerosol-coating of dust" to "secondary coating of dust"

**Minor comment 6:**

*"How different are the control climates between the different models? Would you expect this to impact your results?"*

The control climates are fairly similar between the models – an annual mean climatology is plotted below for comparison. If compared with observations, all three models have similar magnitude temperature biases. GISS is a bit too warm in the tropical oceans, CESM is a bit too warm over the northern mid-latitude land, all three - although especially HadGEM - are too warm in the Southern Ocean, and possibly too cold over the polar regions, by a few degrees in each case. On average, GISS is about ½ a degree warmer than HadGEM, which is about ½ a degree warmer than CESM.

In terms of whether this would impact our results – we do not think it could have a substantial difference to the models' responses to an aerosol emissions perturbation. Firstly, because the changes in SW flux themselves explain much of the diversity in the models' temperature responses. The effect of climate feedbacks and climate sensitivity may play a role in setting the exact magnitude of the final response, but these vary between models anyway, unrelated to the climate state, and so this is part of the structural uncertainty we wish to explore. To our knowledge, studies that have looked for example at the time-dependence of climate sensitivity and feedbacks in transient warming scenarios generally find that it varies slowly, and so inter-model variations in climate sensitivity are likely much more important than the base climate state, unless this were to be very different.

[Figure]

**Minor comment 7:**

"*P7L27: Are the runs restarted from an earlier simulation?  50-year spin-up by itself doesn't seem sufficient for a coupled model.*"

Yes, the runs were restarted from previous coupled simulations that had already been run for present-day conditions, though not necessarily with the exact model set-up that was used here.  The 50 years is not intended to spin-up the control runs, but rather to allow the response to the perturbation to establish itself.  We have expanded the experimental setup section to clarify this.  Previous studies which apply an abrupt forcing (e.g. Andrews et al. (GRL, 2012, doi:10.1029/2012GL051942) have generally seen that most of the global surface temperature response is realised within this timeframe, and from inspection of the time series of global temperature changes, this seemed to be the case here as well.

Changes made:

1) In Section 2.2 (Experimental Setup), added ", initialised from a present-day state," to the description of the control simulations, and "from the same initial state," to the description of the perturbation simulations.

2) In first sentence of Section 3, replaced bracketed phrase "the first 50 years were discarded as spin-up" with "the first 50 years are discarded to allow the response to the perturbation to establish itself".

**Minor comment 8:**

"*P10: Both HadGEM and CESM1 simulate H2O2 and O3 oxidation pathways in the aqueous phase, so including both pathways cannot be an explanation to fast conversion to SO4 in HadGEM. This should be explicitly stated.*"

We agree that including both pathways cannot explain any differences in the $SO_2$ oxidation rates between HadGEM3 and CESM1, only for HadGEM3 and GISS-E2.  We have added a sentence explicitly stating this.

Changes made:

1) Added additional sentence at the start of fourth paragraph of Section 4.1:

"CESM1 includes the same oxidation pathways as HadGEM3-GA4, and in fact has a slightly shorter SO2 lifetime still, and so the differences between these two models have different origins."

**Minor comment 9:**

"*P16L6-8:  Do you refer to sulphate aerosol above cloud top here?  Simulated cloud distributions can have large impacts also in other ways, e.g. the background aerosol amount (clean/polluted) has large impacts on indirect effects, which start to saturate at high aerosol concentrations.*"

Yes – as suggested we've rephrased this sentence to add a mention here of other ways cloud distribution has a potential impact via the saturation of indirect effects (plus reference).

Changes made:

1) Added to first paragraph of Section 4.2 (section in square brackets was already there):

    "For instance, [the radiative effect of sulfate aerosol is modulated by the reflectivity of the underlying surface in the radiation scheme (Chýlek and Coakley, 1974; Chand et al., 2009), which may often be a cloud-top.]  The low contrast with a highly reflective cloud surface means that sulfate aerosol above a cloud top will have a reduced direct radiative forcing.  Blocking of radiation by clouds will also reduce the direct radiative effects of any aerosols within or below them (e.g. Keil and Haywood, 2003).  Additionally, aerosol indirect effects can saturate in regions with a high level of background aerosol (e.g. Verheggen et al., 2007; Carslaw et al., 2013), meaning that the potential for indirect radiative forcing can also vary with the location of clouds.  On top of diversity in indirect effects, and in the climatological distribution of clouds, different dynamical changes in cloud cover could also alter the all-sky flux."

2) Inserted additional references (Keil and Haywood, 2013, Verheggen et al., 2007, and Carslaw et al., 2013) in bibliography

**Minor comment 10:**

*"P16L20-21: Can you speculate which dynamical processes cause the increase in cloudiness when sulphate is removed? Based on 2nd indirect effect one would assume decreased cloudiness."*

Indeed, we also expected decreased cloudiness from the 2nd indirect effect, and so the observed increases in GISS are presumably dynamical in origin.  Dynamical feedbacks can be complex and chaotic, and cause and effect hard to untangle.  Moreover, despite all having local warming at the surface in east China, all three models have quite different regional cloud changes, and so whatever dynamical processes are at play are not robust.   Therefore, we do not wish to speculate further here.

---

## Author Comment (AC4) · 12 Jun 2016

**"Regional and global climate response to anthropogenic SO2 emissions from China in three climate models" by M. Kasoar et al.**

**Additional changes:**

1) Added "cloud radiative interactions" to list of key discrepancies in the abstract

2) Added Boucher et al. (2013) reference to the overview of aerosol radiative effects in the Introduction

3) Added Meinshausen et al. (2011) reference for the CMIP5 greenhouse gas concentrations used in the model description (Section 2.1)

4) Added additional paragraph to the Experimental Setup (Section 2.2):

   "Additionally, shorter atmosphere-only simulations were performed with HadGEM3-GA4 (identical in setup except that sea-surface temperatures (SSTs) and sea-ice cover are prescribed to year-2000 values) to diagnose the effective radiative forcing, as well as the $SO_2$ oxidation rates and $SO_4$ wet deposition rates for this model, referred to in Section 3, Section 4.1, and Section 4.1.1.  In CESM1, the $SO_2$ burden, surface $SO_4$ concentration, clear-sky radiative flux, and cloud cover referred to in Sections 4.1.1, 4.2, and 4.3, were all diagnosed from a 30-year extension of the control and perturbation coupled simulations, rather than from the original 200 years."

   Removed phrase "where sea-surface temperatures (SSTs) and sea-ice cover were prescribed to year-2000 values" where it originally occurred later on in 4th paragraph of Section 3.

5) HadGEM3-GA4 plots for SW flux change and surface temperature change have been replotted to fix an error in the location of a small number of the significance stipples.  The discussion of the plots is unaffected.

6) HadGEM3-GA4 plots for surface air temperature changed to show 1.5m temperature anomaly rather than surface temperature, as this is probably more consistent with the other model's surface air temperature diagnostics.  The (Ch0 – Con) change is almost identical though, and the discussion is not affected (except that global mean temperature changes from 0.114 K to 0.115 K))

7) Changed name of Section 4.1 from "Differences in simulated aerosol amounts" to "Differences in simulated aerosol amounts and aerosol optical depths"

8) Added clarification at start of 2nd paragraph of Section 4.1 of the source of chemistry diagnostics:

   "For GISS-E2 and HadGEM3-GA4, more detailed chemistry diagnostics were available from a 5-year period of a HadGEM3-GA4 atmosphere-only control simulation, and a 5-year period of the GISS-E2 coupled control simulation.  For these two models,…"

9) Added new penultimate paragraph to Section 4.1 (and references therein):

"The AOD changes per unit burden change are summarised in Table 2, and it is clear that there is a large diversity between the models. The possible contributors to diversity in the AOD per unit burden are extensive, and a full analysis of them is beyond the scope of this paper. Host model effects, such as different cloud climatologies and radiative transfer schemes, are one likely contributor. Stier et al. (2013) suggests that one third of total diversity originates there. Relative humidity, which drives water uptake (hygroscopic growth), is also diverse among models. For example, Pan et al. (2015) find that over India, boundary-layer RH is the main source of diversity. At the more basic level, assumed composition and hygroscopic growth curves also often differ between models – in this case, the aerosol scheme used for HadGEM3-GA4 assumes that all sulfate is in the form of ammonium sulfate, whereas CESM1 and GISS-E2 both assume a mixture of ammonium sulfate and sulfuric acid, and additionally all three models use different sources for their hygroscopic growth parameterisations (Bellouin et al., 2011; Liu et al., 2012; Koch et al., 2011; and references therein)."

10) Added CESM1 to Zhang et al. surface $SO_4$ comparison figure and IMPROVE comparison figure (Supplementary figures S5 and S7), and added CESM1 station biases in each case to the text in Section 4.1.1.

11) Added CESM1 to climatological column $SO_4$ figure (Supplementary Figure S6)

12) OMI $SO_2$: Added an extra Supplementary Figure (S8), to additionally compare column $SO_2$ in GISS-E2, CESM1, and HadGEM3-GA4 with satellite observations from the Ozone Monitoring Instrument (OMI). Split the 2$^{nd}$ last paragraph in Section 4.1.1 (dealing with wet deposition observations) into two, in order to insert a short paragraph about OMI $SO_2$ as follows:

"…Returning to Asia, we therefore also tried evaluating the models against column sulphur dioxide observations. We use the gridded, monthly mean Level 3 observations from the Ozone Monitoring Instrument (OMI) (Krotkov et al, 2008) (available from http://disc.sci.gsfc.nasa.gov/Aura) which is flown on the Aura satellite, averaged over eight years from 2005 - 2012. Over the E. China region the mean OMI $SO_2$ is 0.153 Dobson Units (DU), and all three models appear to overestimate this substantially, with very similar regional mean $SO_2$ columns of 0.282 DU for HadGEM3-GA4, 0.272 DU for GISS-E2, and 0.259 DU for CESM1. Spatially, all three models have more diffuse $SO_2$ fields than the OMI observations, where the $SO_2$ burden seems much more localised around source regions (Supplementary Fig. S8). This may be partly due to the coarse resolution of the models compared with the 0.25°satellite product, but also suggests that the lifetimes for SO2 may be too long in both models, or transport processes too efficient. The surprisingly similar column $SO_2$ burdens in all three models suggests that, at least on regional scales, column $SO_2$ may not constrain $SO_4$ burden that well.

An alternative observational measure which to an extent reflects a column-integrated quantity is the deposition rate, and for the two extreme cases of HadGEM3-GA4 and GISS-E2 we therefore also try comparing against observations of sulfate wet deposition. We use the 3-year mean wet deposition data from 2000-2002 described in Vet et al. (2014)…"

13) Clear-sky SW flux data for GISS-E2 replaced with data from a different clear-sky diagnostic which should be more comparable to the way this variable is calculated in the HadGEM3-

GA4 and CESM1 diagnostics.  HadGEM3-GA4 clear-sky data in Table 2 updated (was previously diagnosed at the surface, now diagnosed at the TOA to be consistent with all-sky diagnostics).  Also updated GISS-E2 and HadGEM3-GA4 clear-sky and all-sky SW flux changes in Figure S10 to show TOA flux changes, using the updated GISS diagnostic.

14) Third and fourth paragraphs of Section 4.2 largely re-written to reflect new GISS-E2 clear-sky SW diagnostic, which points to a larger role for cloud interactions in reducing the sulfate radiative forcing in this model.  They now read:

"For the extreme cases of HadGEM3-GA4 and GISS-E2, comparing the changes in clear-sky TOA SW flux with the all-sky TOA SW flux anomalies (Table 2 and Supplementary Fig. S10) reveals that for clear-sky conditions, there is in fact a much smaller regional discrepancy between these two models: Over the E. Asia region GISS-E2 has a 4.1 Wm-2 clear-sky SW flux change, whereas HadGEM3-GA4 has a 5.1 Wm-2 flux change.  HadGEM3-GA4 still has the larger radiative change, but nowhere near the 6-fold difference that is seen in the all-sky flux (Section 3, and Table 2).  This much reduced difference between GISS-E2 and HadGEM3-GA4 in the clear-sky compared with all-sky anomaly is hard to apportion quantitatively though, because compared with the clear-sky change, the all-sky response incorporates all the contributing factors described above:   the additional radiative forcing due to aerosol indirect effects, the screening of direct radiative effects due to clouds blocking radiation and providing a high albedo background, and also any dynamical changes in cloud cover.

In this case, GISS-E2 is found to simulate a small increase in cloudiness in east China due to dynamical changes when sulfate is removed (Supplementary Fig. S11a).  Combined with the screening effect of clouds, this appears to almost completely offset the direct forcing of reduced SO4, and results in a far smaller all-sky flux change than clear-sky flux change over E. China (0.9 Wm-2 all-sky compared with 4.1 Wm-2 clear-sky).  HadGEM3-GA4 by contrast has very little difference between all-sky and clear-sky flux changes (5.3 Wm-2 and 5.1 Wm-2 respectively (Table 2)).  The changes in cloud amount over east China are somewhat more mixed (Supplementary Fig. S11c), though area-averaged, the overall cloud change is a small decrease, which should enhance the all-sky flux change.  However, spatially as well as in magnitude the HadGEM3-GA4 all-sky flux change is exceptionally similar to its clear-sky flux change, and does not resemble the pattern of cloud changes (comparing Supplementary Figs. S10e,f, and Fig. S11c), which suggests that aerosol radiative effects are larger than the effect of the small cloud cover changes, and still dominate the all-sky flux changes. Therefore, the very similar regional all-sky and clear-sky SW flux changes in HadGEM3-GA4 implies that unlike in GISS-E2, aerosol indirect effects in HadGEM3-GA4 probably roughly compensate for the presence of clouds reducing the direct effect, so that the change in all-sky combined direct and indirect forcing is similar to the change in clear-sky direct forcing when sulfate is removed."

15) Added additional Supplementary Figure (S11) showing regional cloud cover changes in all three models (referred to in Section 4.2 where it previously said 'not shown')

16) Added additional caveat to end of Section 4.2:

"We note though that clear-sky diagnostics will be influenced by choices within the models of how aerosol water uptake is determined under the artificial assumption of clear-sky

conditions.  The all-sky SW flux change, which drives the final climate response, is regionally still the most directly comparable quantity, reflecting the total radiative effect of the aerosol change."

17) In second paragraph of Section 4.3, added/changed marked-up text in following sentence:

"This is not directly comparable with previous studies like Myhre et al. (2013a), as we use a regionally-averaged number instead of globally-averaged, and for the numerator we use the change in clear-sky TOA SW flux as the best available measure of aerosol direct radiative effect, rather than the  direct radiative forcing diagnosed either from double radiation calls or simulations with fixed meteorology.

18) In third paragraph of Section 4.3, changed regional radiative forcing efficiency values for HadGEM3-GA4 and GISS-E2 to reflect new diagnostics used in Table 2.  This results in GISS-E2 have a much higher value than HadGEM3-GA4, rather than just a somewhat higher value. It also changes the flux change normalised by sulfate burden change so that GISS-E2 is now bigger than HadGEM3-GA4, rather than smaller.  Relevant comparative statements in this paragraph were therefore changed as shown in mark-up:

"As noted in Sect. 4.1 and 4.2, over the eastern China region HadGEM3-GA4 has a 6-fold larger mean AOD reduction (-0.29) compared with GISS-E2 (-0.047), but only slightly larger clear-sky SW change (5.1 W m-2 compared with 4.1 W m-2).  As a result, the regional radiative efficiency for HadGEM3-GA4 is much smaller than that of GISS-E2: (-17.6 W m-2 compared with -87.2 W m-2) per unit AOD change (Table 2). If instead of AOD we normalise by the change in sulfate burden  integrated over the same region we find a similar relationship: HadGEM3-GA4 has a smaller regional mean change in clear-sky SW flux per Tg sulfate than GISS-E2: (-145 W m-2 Tg-1 compared with -256 W m-2 Tg-1).  Proportionally though, the discrepancy is not as great when normalising by change in sulfate burden, due to the  much larger AOD per unit mass of sulfate simulated in HadGEM3-GA4 .  Curiously Myhre et al. (2013a) reported results that were qualitatively the inverse of what we show here, finding that the atmospheric component of GISS ModelE has a smaller sulfate radiative forcing than that of HadGEM2 (HadGEM3's predecessor, with a very similar aerosol scheme) when normalised by AOD, although still larger when normalised by column-integrated sulfate burden.

19) In final paragraph of Seciton 4.3, inserted the word "all-sky" into sentence:

"In their case, they found CAM5.1 to have approximately 2.25 times higher all-sky direct radiative forcing per unit AOD than GISS-E2."

20) In Section 4.4, replaced (Samset et al., in preparation) with (Samset et al., 2016) and updated reference in the reference list, since this paper is now published.

21) Last paragraph of Section 4.4, removed four instances of the word 'regional' when referring to the Shindell (2012) study which looked at forcings imposed in different latitude bands, to avoid confusing with the more localised usage of 'regional' throughout the rest of the paper to refer to the China /East Asia region.  In the last sentence of this section, replaced 'regional

forcings' with 'forcings at different latitudes'.

22) Other minor grammatical and readability changes – see tracked changes in full manuscript for details